# Altered temporal sequence of transcriptional regulators in the generation of human cerebellar granule cells

**Hourinaz Behesti[1], Arif Kocabas[1], David E Buchholz[1], Thomas S Carroll[2], Mary E Hatten[1]***

[1]Laboratory of Developmental Neurobiology, Rockefeller University, New York, United States; [2]Bioinformatics Resource Center, Rockefeller University, New York, United States

**Abstract** Brain development is regulated by conserved transcriptional programs across species, but little is known about the divergent mechanisms that create species-specific characteristics. Among brain regions, human cerebellar histogenesis differs in complexity compared with nonhuman primates and rodents, making it important to develop methods to generate human cerebellar neurons that closely resemble those in the developing human cerebellum. We report a rapid protocol for the derivation of the human ATOH1 lineage, the precursor of excitatory cerebellar neurons, from human pluripotent stem cells (hPSCs). Upon transplantation into juvenile mice, hPSC-derived cerebellar granule cells migrated along glial fibers and integrated into the cerebellar cortex. By Translational Ribosome Affinity Purification-seq, we identified an unexpected temporal shift in the expression of RBFOX3 (NeuN) and NEUROD1, which are classically associated with differentiated neurons, in the human outer external granule layer. This molecular divergence may enable the protracted development of the human cerebellum compared to mice.

*For correspondence:
hatten@rockefeller.edu

## Editor's evaluation

Your paper beautifully addresses the divergent mechanisms that create species-specific features of brain development, focusing on transcriptional programs and their timing, that generate human cerebellar cells compared to those in mouse. You describe a rapid protocol for the derivation of the human ATOH1 lineage that generates excitatory cerebellar neurons from human embryonic stem cells (hESCs), and study them in vitro and in vivo. You observed transcription factors classically associated with mouse differentiated neurons expressed in the human outer external granule layer where granule cell precursors reside. These results argue that the prolonged development of the cerebellum in the human is linked to its increased size in evolution.

## Introduction

Understanding the development of the human brain is an emerging area of neuroscience. The human cerebellum is now recognized to contribute to cognitive functions (*Allen et al., 1997*; *Carta et al., 2019*; *Fiez, 1996*; *Schmahmann et al., 2019*; *Stoodley et al., 2017*; *Wagner et al., 2017*), in addition to a critical role in motor control and motor learning. As one of the most ancient cortical regions, the cerebellum also appears central to human cognitive evolution, having rapidly expanded in absolute size, and relative to the neocortex in apes and humans (*Barton and Venditti, 2014*). Recent studies

provide evidence of the complexity of human cerebellar histogenesis compared with nonhuman primates and rodents (*Haldipur et al., 2019*), making it important to develop methods to generate human cerebellar neurons to model human development and disease. The cerebellar cortex develops from rhombomere 1 (*Wingate and Hatten, 1999*) with a primary germinal zone that produces the Purkinje neurons and interneurons, and a secondary germinal zone, marked by the *ATOH1* transcription factor, that emerges from the rhombic lip and generates cerebellar granule cells (GCs) (*Hatten and Heintz, 1995*). Importantly, prior to the specification of granule cell progenitors (GCPs), the ATOH1 lineage gives rise to a subset of hindbrain nuclei and cerebellar nuclei in mice (*Machold and Fishell, 2005*; *Wang et al., 2005*). While previous studies have mostly focused on generating human Purkinje cells (*Buchholz et al., 2020*; *Erceg et al., 2010*; *Muguruma et al., 2015*; *Nayler et al., 2017*; *Silva et al., 2020*; *Wang et al., 2015*; *Watson et al., 2018*) from human pluripotent stem cells (hPSCs), little attention has been given to defining the molecular pathways that generate human GCs and other rhombic lip derivatives. The importance of understanding the human ATOH1 lineage is underscored by the fact that GCPs are a known cell of origin for medulloblastoma, the most common metastatic childhood brain tumor (*Behesti and Marino, 2009*; *Marino et al., 2000*), and GCs are implicated in neurodevelopmental disorders including autism (*Bauman, 1991*; *Kloth et al., 2015*; *Menashe et al., 2013*).

Here, we report a rapid and simple protocol for the directed derivation of the human ATOH1 lineage, the precursor of excitatory cerebellar neurons, by the sequential addition of six factors, in a chemically defined culture medium. Using transgenic reporter lines and Translational Ribosome Affinity Purification (TRAP)-seq adapted to hPSCs, we tracked developmental gene expression in culture in a lineage-specific manner. Compared to previously reported studies, this method accelerates the production of $ATOH1^+$ cells (day in vitro [DIV] 16 versus DIV35) in previous studies (*Muguruma et al., 2015*), with a dramatic increase in yield (80% vs. 17%). Strategies to overcome culture variability in terms of gene expression, a common limitation of hPSC-derived models, included the addition of BMP7 to stabilize *ATOH1* gene expression, patterning of cells from a single-cell stage, and growth on transwell membranes where cells have access to medium in two dimensions (above and below). The translational profile of the hPSC-derived ATOH1 lineage most closely resembled human cerebellar tissue in the second trimester compared to other brain regions. Finally, we report the discovery of a shift in the expression of transcriptional regulators (RBFOX3 [NeuN] and NEUROD1) in the progenitor zone of the human external granule cell layer (EGL). NeuN and to a large extent NeuroD1 are expressed in postmitotic neurons in vertebrates (*Miyata et al., 1999*; *Mullen et al., 1992*). This molecular divergence may provide the mechanism whereby the GCP pool persists into year 2 post birth in humans, but only for 2 weeks in mice.

## Results

### Directed derivation of the human ATOH1 lineage from hPSCs

Addition of dual SMAD inhibitors for neuralization (*Chambers et al., 2009*), followed by the addition of fibroblast growth factor (FGF) and a small-molecule agonist of WNT signaling (CHIR99021, 'CHIR' from hereon) for posteriorization, in a chemically defined serum-free medium, induced the expression of anterior hindbrain markers *EN2*, *MEIS2*, *GBX2* and repressed midbrain (*OTX2*), and spinal cord-level (*HOXA2*) markers (*Figure 1—figure supplement 1A*) by DIV11. FGF + CHIR treatment was superior at inducing mid/hindbrain markers (*EN2*, *GBX2*) and reducing mid/forebrain markers (*OTX2/PAX6*) compared to a previously reported combination of insulin + FGF2 (*Muguruma et al., 2015*; *Figure 1—figure supplement 1A*). Empirical testing of the timing, duration, and concentrations of CHIR + FGF, necessary for a 'cerebellar territory' expression profile of $EN2^+$;$GBX2^+$;$OTX2^-$, revealed that the addition of 2.5 μM CHIR from DIV1 until at least DIV9–11 (*Figure 1A*, *Figure 1—figure supplement 1B*) was necessary and that dual SMAD inhibition until DIV7 was sufficient (*Figure 1—figure supplement 1B*; data not shown). While both FGF8b (*Chi et al., 2003*; *Guo et al., 2010*; *Martinez et al., 1999*) and FGF2 (*Muguruma et al., 2015*) treatments induced cerebellar territory, FGF2 induced higher *EN2* levels (*Figure 1—figure supplement 1B*) and improved cell survival (*Figure 1—figure supplement 1C*).

Although this method consistently yielded cerebellar territory in two hPSC lines (*Figure 1—figure supplement 1D*), gene expression levels were variable between experiments. To reduce variability,

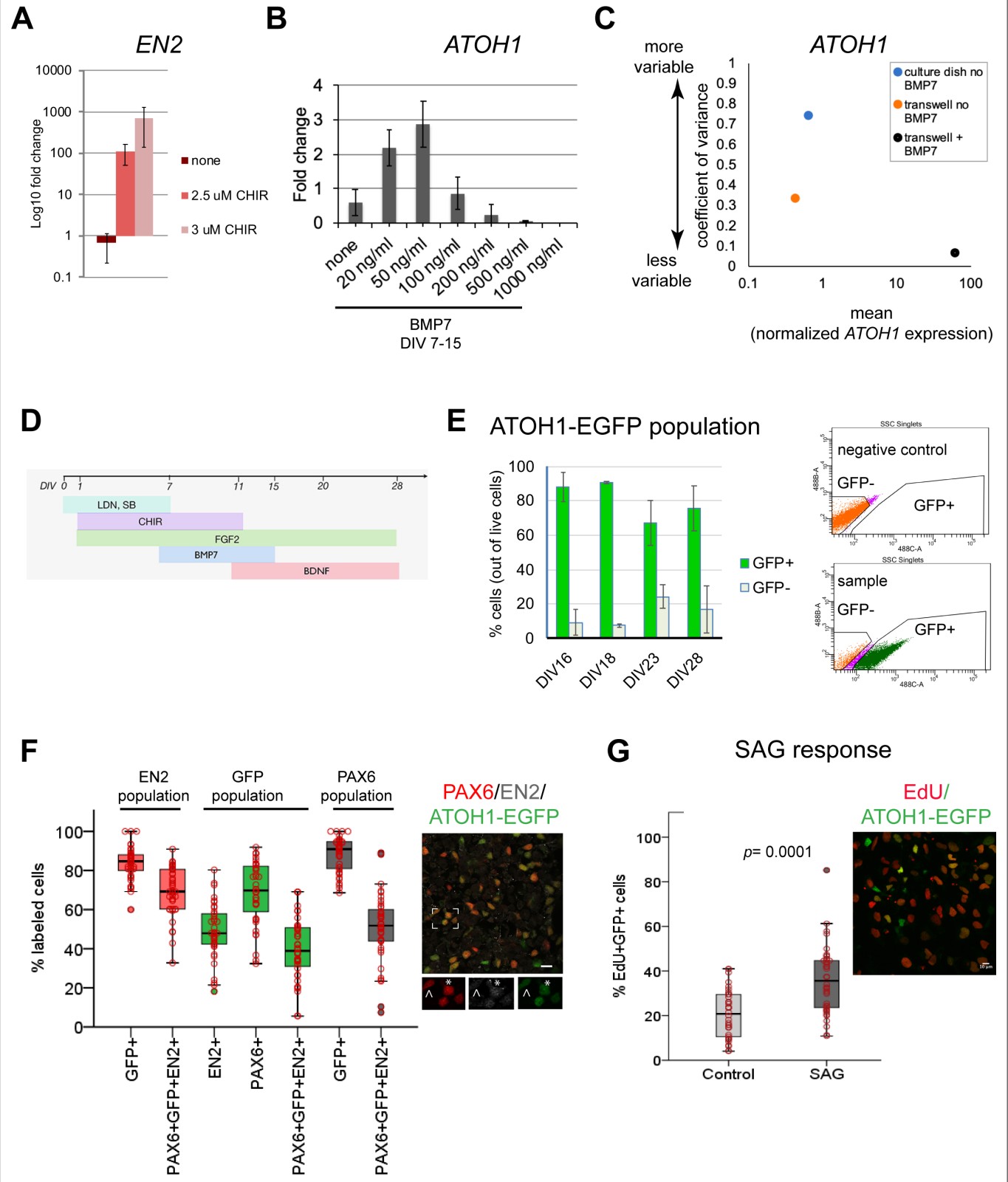

**Figure 1.** Derivation of the human ATOH1 lineage. (**A**) *EN2* expression (log$_{10}$ fold change of no CHIR99021) in dual SMAD+FGF2-treated human pluripotent stem cells (hPSCs) in the absence and presence of CHIR99021 by RT-qPCR at day in vitro (DIV) 11. (**B**) *ATOH1* expression (fold change of no BMP7) at DIV16 in response to a BMP7 concentration series added at DIV7–15. (**C**) Dot plot showing the coefficient of variance of mean *ATOH1* expression detected by RT-qPCR at DIV16 in cultures grown on regular tissue culture dishes (-BMP7, blue) versus on transwell membranes (-BMP7,

*Figure 1 continued on next page*

*Figure 1 continued*

orange; +BMP7, black). (**D**) Schematic of the protocol for derivation of the ATOH1 lineage. (**E**) Left: the percentage of EGFP⁺ (green) and EGFP⁻ (gray) cells at DIV16, 18, 23, and 28 of differentiation of the *ATOH1-EGFP* line by fluorescence-activated cell (FAC)-sorting (the change in EGFP⁺ population across DIVs compared by ANOVA: p=0.053). Right: representative FACS charts showing separation of ATOH1-EGFP⁺/EGFP⁻ cells. (**F**) Left: box plot showing the percentages of EGFP, EN2, PAX6 single- and triple-positive cells by immunocytochemistry, within the EN2, EGPF (ATOH1), and PAX6 populations at DIV28–30. Right: representative merged image of the immunocytochemistry labeling. Boxed area is magnified at the bottom with individual channels displayed. Note asterisk highlighting a triple-positive cell, while cell above the arrowhead is EN2⁻;PAX6⁺;EGFP⁺. (**G**) Left: box plot showing the percentage of EdU⁺;EGFP⁺ double-positive cells per Dapi nuclei ± SAG treatment after 48 hr (DIV28–30). Right: a representative merged image of the labeling. N = 3 independent experiments except in (**B**), which shows technical replicates. Bar graphs show mean ± 1 SD. Scale bars: 10 µm.

The online version of this article includes the following figure supplement(s) for figure 1:

**Figure supplement 1.** Derivation and characterization of the human *ATOH1* lineage.

**Figure supplement 2.** Characterization of *ATOH1-EGFP* and *ATOH1-EGFP-L10a* transgenic lines.

**Figure supplement 3.** Characterization of ATOH1-EGFP cells (until day in vitro [DIV] 28).

a common limitation of 2D and 3D stem cell differentiation methods, we tested different culture surfaces (material and area), and the addition of signaling molecules to override stochastic gene expression. Sparse single-cell plating of hPSCs on transwell membranes allowed the formation of similarly sized and shaped colonies, resembling the growing neural plate, as patterning proceeded (*Figure 1—figure supplement 1E*). This improved the consistency of the geometric arrangement of the patterned cells as early colonies, reduced gene expression variability (*Figure 1C*), and increased cell survival, likely due to cells receiving medium in 2D (above and below) compared to 1D in regular culture dishes.

Previous work has shown that roof plate-derived BMP7 induces *Atoh1* expression during GCP specification in the mouse hindbrain (*Alder et al., 1999*). Without knowledge of the effective concentration of secreted BMP7 in the developing human brain, we tested a range of concentrations and importantly found that low BMP7 concentrations increased *ATOH1* expression, while higher levels, previously used for mouse ESC-derived GCs (*Salero and Hatten, 2007*), induced low levels of *ATOH1* expression and caused cell death (*Figure 1B*; data not shown). BMP7 treatment further reduced *ATOH1* expression variability (*Figure 1C*). Therefore, the addition of BMP7 appears to override the stochasticity of *ATOH1* expression in cultures. Finally, BDNF was added to improve GC survival (*Lindholm et al., 1993*). Together, these experiments defined optimal conditions for deriving a human cerebellar territory and ATOH1 lineage specification as first steps towards derivation of the ATOH1 lineage and GC differentiation (*Figure 1D*).

To monitor the dynamics of ATOH1 lineage specification and differentiation in culture, we derived a clonal *ATOH1-EGFP* hPSC line, expressing nuclear EGFP under a human *ATOH1* enhancer (*Figure 1—figure supplement 2A*). Fluorescence-activated cell (FAC)-sorting of ATOH1-EGFP⁺ cells at DIV16 followed by RT-qPCR revealed coexpression of genes associated with the cerebellar territory in mice (Allan Brain Atlas, *Morales and Hatten, 2006*) including *ATOH1, PAX6, EN2, ID4, LHX2, LHX9*, and *MEIS2* (*Figure 1—figure supplement 1F*) in ATOH1-EGFP⁺ cells. Time-series analysis of ATOH1-EGFP by imaging and flow cytometry revealed that by DIV16, 80 ± 9% of the cells are ATOH1-EGFP⁺ (*Figure 1E*). This is a fivefold increase in the efficiency of ATOH1⁺ cell derivation compared to a previously reported 3D protocol (80 vs. 17%; *Muguruma et al., 2015*). By DIV23, the percentage of ATOH1-EGFP⁺ cells decreased, while ATOH1-EGFP⁻ cells increased, indicating the onset of differentiation (*Figure 1E*). To investigate when the ATOH1-EGFP⁺ cells coexpress known GCP markers, gene expression was analyzed in FAC-sorted ATOH1-EGFP cells at three time points between DIV16-23. In the mouse, the selected genes are dynamically expressed within the *Atoh1*⁺ domain, changing between embryonic day (E) 11.5, when the *Atoh1*⁺ population gives rise to hindbrain and cerebellar nuclei, and E15.5, when the *Atoh1*⁺ population forms the EGL (*Figure 1—figure supplement 1F*, right; *Machold and Fishell, 2005*; *Wang et al., 2005*; Allan Brain Atlas). Consistent with the reduction in ATOH1-EGFP⁺ cells by DIV23, gene expression shifted between DIV19 and 23, with markers expressed in the mouse *Atoh1* domain at E11.5, prior to EGL establishment (*ID4, LHX2, LHX9*), decreasing in expression, while EGL markers increased (*PAX6, ATOH1*) (*Figure 1—figure supplement 1F*). Thus, at DIV23 the hPSC-ATOH1 lineage initiates GCP production but likely contains a mixture of progenitors. Indeed, prior to DIV23, at DIV16, a small subset of cells expressed Calretinin by immunohistochemistry (*Figure 1—figure supplement 3A*), a marker of excitatory cerebellar nuclei

produced by the Atoh1 lineage prior to GC production. By immunohistochemistry, at DIV28, the ATOH1[+] cells coexpressed PAX6 (mean ~70%) and EN2 (mean ~50%), and ~40% coexpressed all three markers within the EGFP[+] population. The percentage of coexpression of these three markers within the EN2[+] (marker of mid/hindbrain) population was ~70% (*Figure 1F*). Thus, a great majority of the EN2[+] cells are GCPs. Midbrain progenitors that express EN2 do not express ATOH1/PAX6 (*Akazawa et al., 1995*; Allan Brain Atlas), and dorsal posterior hindbrain/spinal cord *Atoh1[+]* never express EN2 (*Davidson et al., 1988*).

To examine the identities of cells that are negative for ATOH1-EGP, we performed double immuno-labeling with several other markers at DIV28. These analyses revealed the presence of ATOH1-EGFP[-];PAX6[+],NeuN[+] cells (*Figure 1—figure supplement 3B*, asterisk), indicative of differentiated GCs. A small number of Calretinin[+] cells, indicative of cerebellar nuclei or unipolar brush cells (*Abbott and Jacobowitz, 1995*; *Fujita et al., 2020*; *Wizeman et al., 2019*), and ATOH1-EGFP[-];SOX2[+] cells (28 ± 16% [mean ± 1 SD of the SOX2[+] cells, n = 2]), indicative of cells deriving from the ventricular zone neuroepithelium in the developing cerebellum (GABAergic lineage, *Figure 1—figure supplement 3C*). Interestingly, a great majority of the SOX2[+] cells were ATOH1-EGFP[+] (72 ± 16.7% [mean ±1 SD, n = 2]; *Figure 1—figure supplement 3C and D*). In the developing mouse cerebellum, SOX2 is abundantly expressed throughout the ventricular zone neuroepithelium, but is a rare find in the EGL (*Sutter et al., 2010*).

A defining feature of GCPs is their extensive proliferative capacity in response to sonic hedgehog (SHH) in the postnatal mouse cerebellum (*Dahmane and Ruiz i Altaba, 1999*; *Lewis et al., 2004*; *Wallace, 1999*; *Wechsler-Reya and Scott, 1999*). EdU uptake, a measure of cell proliferation, was significantly increased in the ATOH1-EGFP[+] cells (*Figure 1G*) after treatment with SAG (an agonist of SHH signaling) compared to control. In conclusion, the hPSC-ATOH1[+] lineage displayed similar dynamics in gene expression over time as seen in the mouse embryo (albeit extended in period) and produced GCPs, which respond to SAG, by DIV23–28. Compared to previously reported methods, both the speed (35 days versus 16 days) and the yield (from 17% to 80%) of GCP production were increased.

## Differentiation of hPSCs into cerebellar granule cells

In the mouse, early postmitotic GCs switch off *Atoh1* and transiently express TAG1 in the inner EGL. In our cultures, TAG1 was expressed already at DIV18, after which its expression increased (*Figure 2A*). Magnetic-activated cell sorting (MACS) using an antibody against TAG1, which is expressed on the cell surface, yielded 13 ± 7% (n = 7) TAG1[+] cells at DIV28 (*Figure 2B and C*). TAG1[+] cells were then co-cultured with either mixed cerebellar mouse neurons or glia to provide a permissive differentiation environment. By DIV3 in co-culture, the majority of the TAG1[+] cells were PAX6[+] (*Figure 2D*, top panel). By contrast, the TAG1[-] fraction contained mostly cells with larger nuclei that were PAX6[-] (*Figure 2D*, bottom panel). By DIV20, co-culture (DIV48 total days) of TAG1[+] cells with mouse neurons or glia resulted in a majority of the human cells displaying characteristic GC morphology, namely a small (<10 um) round nucleus and a limited number of extended processes (3–4) including bifurcated axons (*Figure 2E*, 136/230 examined, 60%). Mouse GCs isolated at P0 and grown concurrently in the same dish as the human cells displayed similar features (*Figure 2E*). The human cells expressed the GC markers NEUROD1 (42/56 examined, 75%, *Figure 2E*) and PAX6 (174/218 examined, 80%). In double-labeling experiments, all examined cells that expressed PAX6 (18/22, 81%) also expressed NEUROD1. Interestingly, presynaptic specializations were apparent in neurons grown in co-culture with mouse glia only, indicating that cross-species neuron-glia interactions are sufficient to trigger synapse initiation (*Figure 2F*). In addition, the expression of vesicular glutamate receptor 1 (VGLUT1), a presynaptic marker that in the cerebellum is localized to GC parallel fiber-Purkinje cell excitatory synapses, was detected (*Figure 2G*).

While we carried out the most comprehensive characterization of the MACsorted cells at DIV28 and their differentiation in co-culture with mouse cells, beyond DIV28, it should be noted that the human cultures continued to express ATOH1-EGFP. TAG1 came on transiently in subsets of cells also at later stages (~13% at DIV35; data not shown). At DIV28 + 20, the TAG1[-] fraction contained cells with several morphologies (*Figure 2—figure supplement 1*), including cells with small round nuclei that expressed NEUROD1 (~47%) and NeuN (~70%). A small fraction was Calretinin[+] (~5%). The majority of the Calretinin[+] cells at this stage (70 ± 36% of the 5%) were double positive for NeuN.

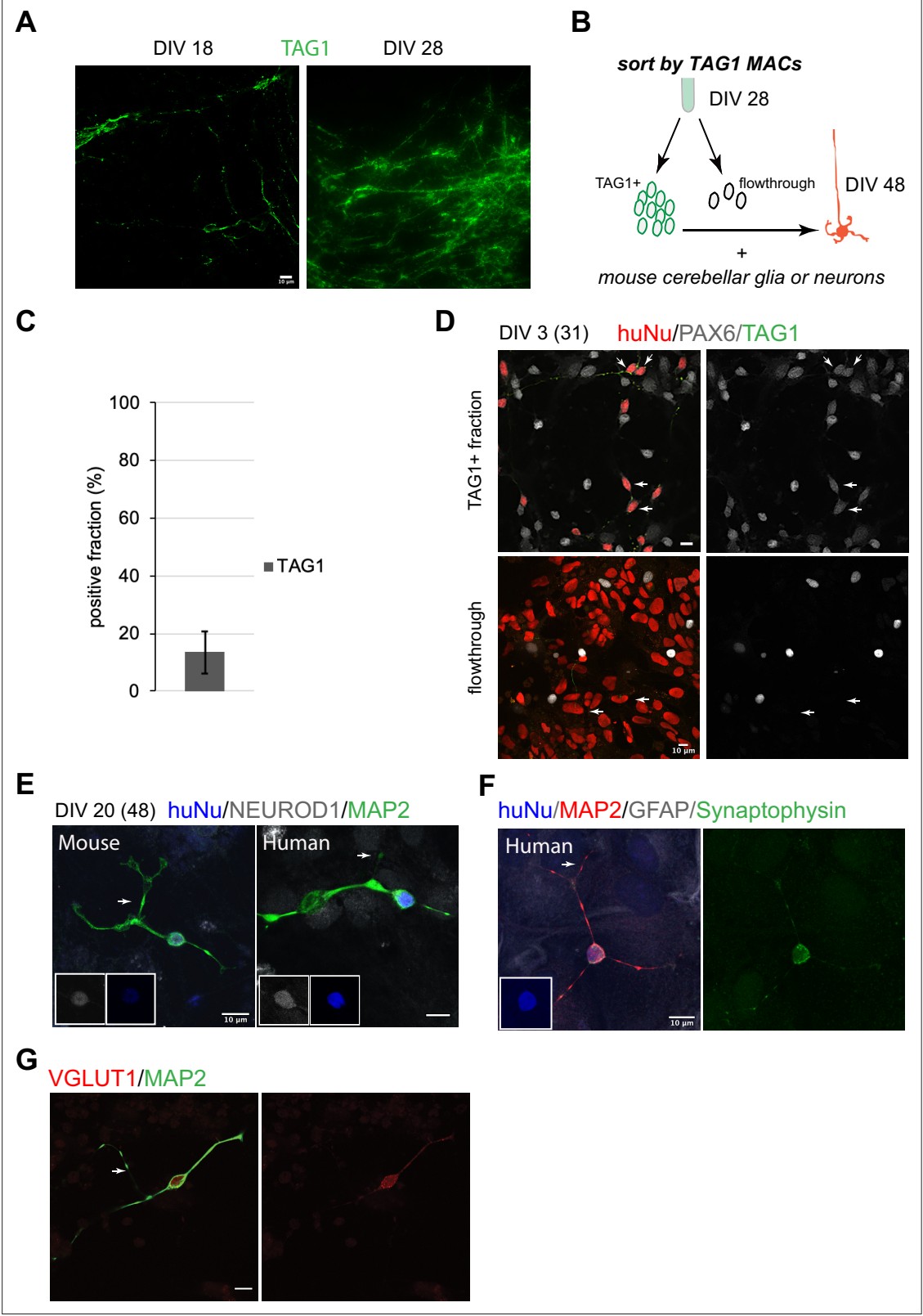

**Figure 2.** Human granule cell (GC) differentiation from human pluripotent stem cells (hPSCs). (**A**) TAG1 expression at day in vitro [DIV]18 (left) and DIV28 (right). (**B**) Schematic of the sorting strategy of TAG1+ cells by magnetic-activated cell sorting (MACS) at DIV28 and co-culture with mouse cerebellar neurons or glia until DIV48. (**C**) Bar chart (mean ± 1 SD) of TAG1+ cells/total cells at DIV28, N = 7 independent experiments. (**D**) Top: TAG1+ cells (green) in co-culture with mouse cerebellar neurons and glia for 3 days express PAX6 (red + white, arrows). Bottom: TAG1- cells (flowthrough) in co-culture with

*Figure 2 continued on next page*

*Figure 2 continued*

mouse neurons and glia have larger nuclei and are PAX6 (arrows). (**E**) TAG1+ cells in co-culture with mouse neurons and glia for 20 days (DIV48 total) display small round nuclei (inset, blue), bifurcated neuronal extensions, and are NEUROD1+;MAP2+. A mouse GC cultured in the same dish for the same period of time, shown for comparison. (**F**) TAG1+ cell in co-culture with glia only for 20 days (DIV48 total) expresses synaptophysin. (**G**) TAG1+ cell in co-culture with mouse neurons and glia (DIV48 total) expresses VGLUT1.

The online version of this article includes the following figure supplement(s) for figure 2:

**Figure supplement 1.** Characterization of TAG1-negative fraction at day in vitro (DIV)28 + 20.

To assess whether hPSC-GCs would integrate into the mouse cerebellar cortex, especially whether they would undergo the classic glial-guided migration of GCs, we implanted human cells into the juvenile mouse cerebellum at the time when mouse GCs are undergoing migration. Importantly, TAG1-sorted cells integrated into the neonatal mouse cerebellar cortex and migrated along glial fibers with stereotypical morphology of migrating neurons passaging through the EGL to settle in the internal

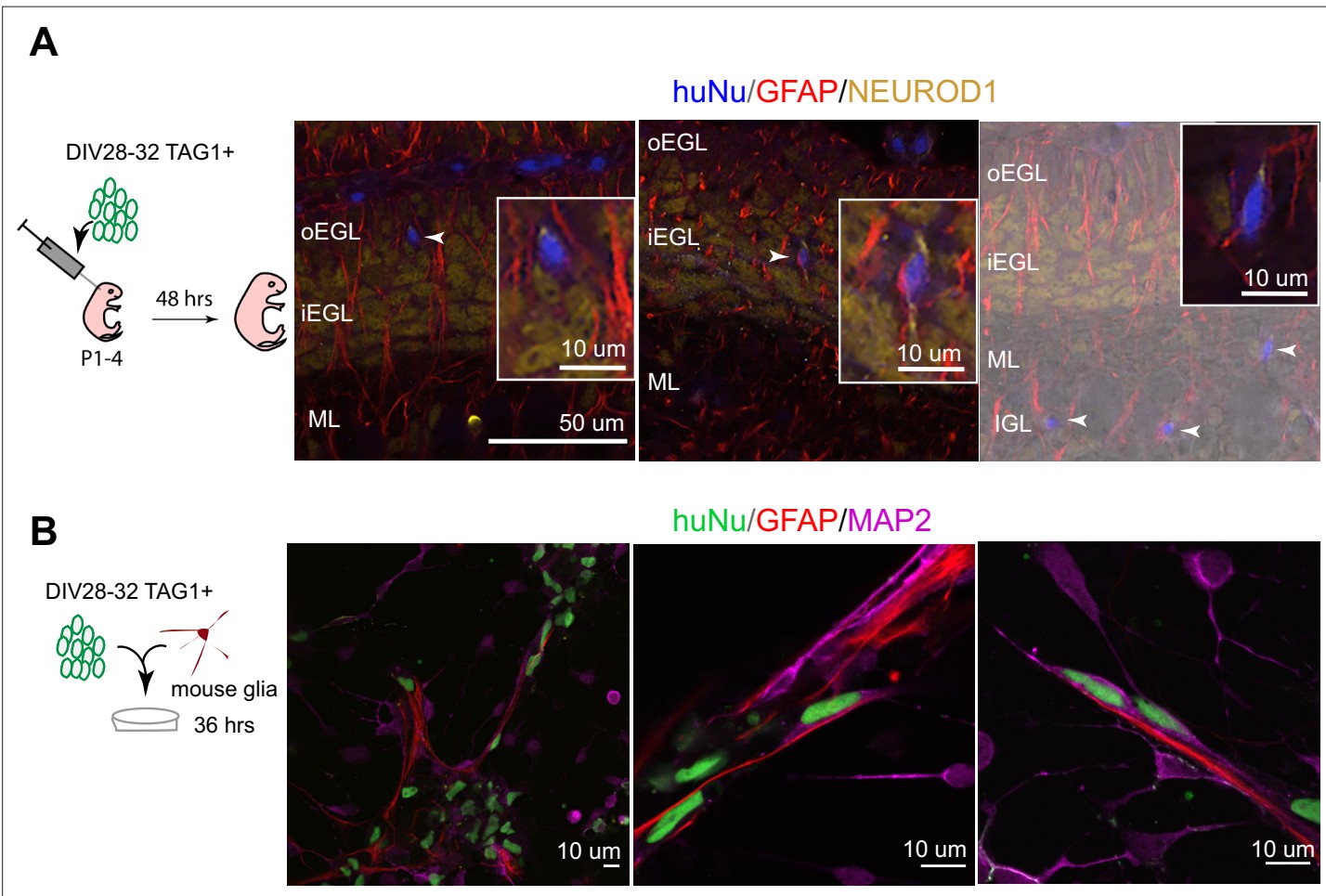

**Figure 3.** Human postmitotic granule cells (GCs) undergo glial-guided neuronal migration and integrate into the mouse cerebellum upon transplantation. (**A**) Left: schematic outlining transplantation of MACsorted (day in vitro [DIV]28–32) TAG1+ human cells into the early postnatal mouse cerebellum. Images show three representative coronal sections of mouse cerebella (N = 5 mice), 48 hr post transplantation. Far-right image is overlaid on a DIC image. Arrowheads highlight human cells integrated in the mouse internal granule cell layer (IGL). Boxed images are higher magnifications of migrating cells. (**B**) Day 28–32 TAG1+ human cells in co-culture with mouse glia after 36 hr, showing examples of migrating neurons with elongated nuclear morphologies along glia. Far-left image shows a lower-magnification view containing migrating neurons on glia as well as non-migrating neurons with rounded morphologies. HuNu, human nuclear antigen; oEGL, outer external granule cell layer; iEGL, inner external granule cell layer; ML, molecular layer. Scale bars as indicated on images.

The online version of this article includes the following figure supplement(s) for figure 3:

**Figure supplement 1.** Integration of human pluripotent stem cell-granule cells (hPSC-GCs) into the mouse cerebellum upon transplantation.

granule cell layer (IGL) (*Figure 3A*, *Figure 3—figure supplement 1*). Moreover, we observed the stereotypical elongated morphology of migrating neurons on glia when TAG1-sorted cells were co-cultured with glia isolated from the mouse cerebellum (*Figure 3B*) for 36 hr, similar to previous observations with mouse GCs (*Hatten, 1985*). By contrast, cells that were not attached to glia displayed more rounded morphologies (*Figure 3B*). Together, these experiments demonstrate that the DIV28 hPSC-GCs can undergo glial-guided migration both in vitro and upon transplantation in vivo.

## hPSC-ATOH1⁺ cells match the molecular profile of the human cerebellum in the second trimester

To transcriptionally profile the hPSC-ATOH1⁺ progenitors, we adapted the TRAP methodology, first developed in transgenic mice for cell-type-specific translational profiling (*Doyle et al., 2008*; *Heiman et al., 2008*), for hPSCs. An EGFP-tagged L10a ribosomal subunit was driven by the human *ATOH1* enhancer (*Figure 1—figure supplement 2B*), enabling GFP-mediated immunoprecipitation (IP) of ribosomally attached mRNAs in the ATOH1 lineage specifically. Importantly, this method bypasses the need for cell dissociation and provides information about transcripts, including the 3′UTR, that are present in both the cell soma and processes. TRAP followed by RNA sequencing (TRAP-seq) was performed at DIV28 when the ATOH1 lineage coexpressed GCP markers and displayed increased proliferation in response to SAG (*Figure 1*). The *ATOH1* transcript was indeed enriched in IPs compared to the input (*Figure 1—figure supplement 2B*, *Figure 4—figure supplement 1A*). Several other GCP markers were similarly enriched, while markers of differentiated GCs were depleted (*Figure 4A*). Heatmaps of the major signaling pathways in development (WNT, BMP, SHH, FGF, HIPPO) highlighted enrichment of the WNT pathway in particular (*Figure 4—figure supplement 1B*). Interestingly, gene ontology analysis revealed axon guidance, the WNT pathway, neuronal migration, and cell division among the top significantly enriched developmental processes (*Supplementary file 1*).

To investigate how the hPSC-ATOH1⁺ cells compare to the molecular signature of the developing human cerebellum, we compared our dataset to RNA-seq data from the PsychEncode study (*Li et al., 2018*), which samples different brain regions during a developmental timeline covering 8 post coitus week (PCW) until after birth in the human. Gene Set Enrichment Analysis (GSEA) revealed that the DIV28 ATOH1 lineage most closely matched the profile of the 13–17 PCW human cerebellum (*Figure 4B and C*, *Figure 4—source data 2*). Moreover, comparisons to published single-cell RNA-seq data from the developing mouse cerebellum revealed most resemblance to various cerebellar glutamatergic lineages (*Figure 4—figure supplement 2*, *Figure 4—figure supplement 2—source data 1*). There was no resemblance to glia or endothelial cells. Together, these analyses provide the first translational dataset for the hPSC-derived ATOH1 lineage. While the data closely match the developing human cerebellum, development appears accelerated in culture.

## Molecular profiling reveals a transcriptional shift in the human versus mouse cerebellum

In a surprising finding, TRAP-seq analysis of the *ATOH1* lineage revealed the expression of several genes that are classically associated with postmitotic GCs in mice including *RBFOX3* (encodes for the NeuN antigen) and *NEUROD1* (*Figure 5A*). The coexpression of *NEUROD1* and *ATOH1* was confirmed by RT-PCR in IPs (*Figure 5A*) and in FAC-sorted EGFP⁺ cells derived from the *ATOH1-EGFP* line (data not shown). By immunohistochemistry, ATOH1-EGFP⁺ cells expressed NeuN and NEUROD1 (*Figure 1—figure supplement 3B*). To investigate if the expression of these factors extends into the proliferative zone of the human EGL in vivo or if this is an in vitro phenomenon, we performed immunohistochemistry in the human cerebellum at 17 PCW as our data most closely matched the human 13–17 PCW. We performed concurrent immunohistochemical analyses on P0 mouse cerebella because the 17 PCW human cerebellum most closely resembles P0 in mice, based on cerebellar foliation depth and pattern (*Biran et al., 2012*; *Haldipur et al., 2019*). In contrast to the mouse, many outer EGL cells in the human, where the *ATOH1*⁺ progenitors reside (*Haldipur et al., 2019*), expressed both NEUROD1 (*Figure 5B and D*, *Figure 5—figure supplement 1A*) and NeuN (*Figure 5C and D*). Moreover, Ki67, a marker of proliferating cells, was not as prevalent at this stage in the human EGL as it is in the P0 mouse cerebellum (*Figure 5B*, high magnification, and C). In the human EGL, Ki67 expression was sparser, Ki67 and NEUROD1 coexpression was clearly evident, and there was no clear separation of cells into two layers as seen in the mouse. Extensive examination of multiple stages of

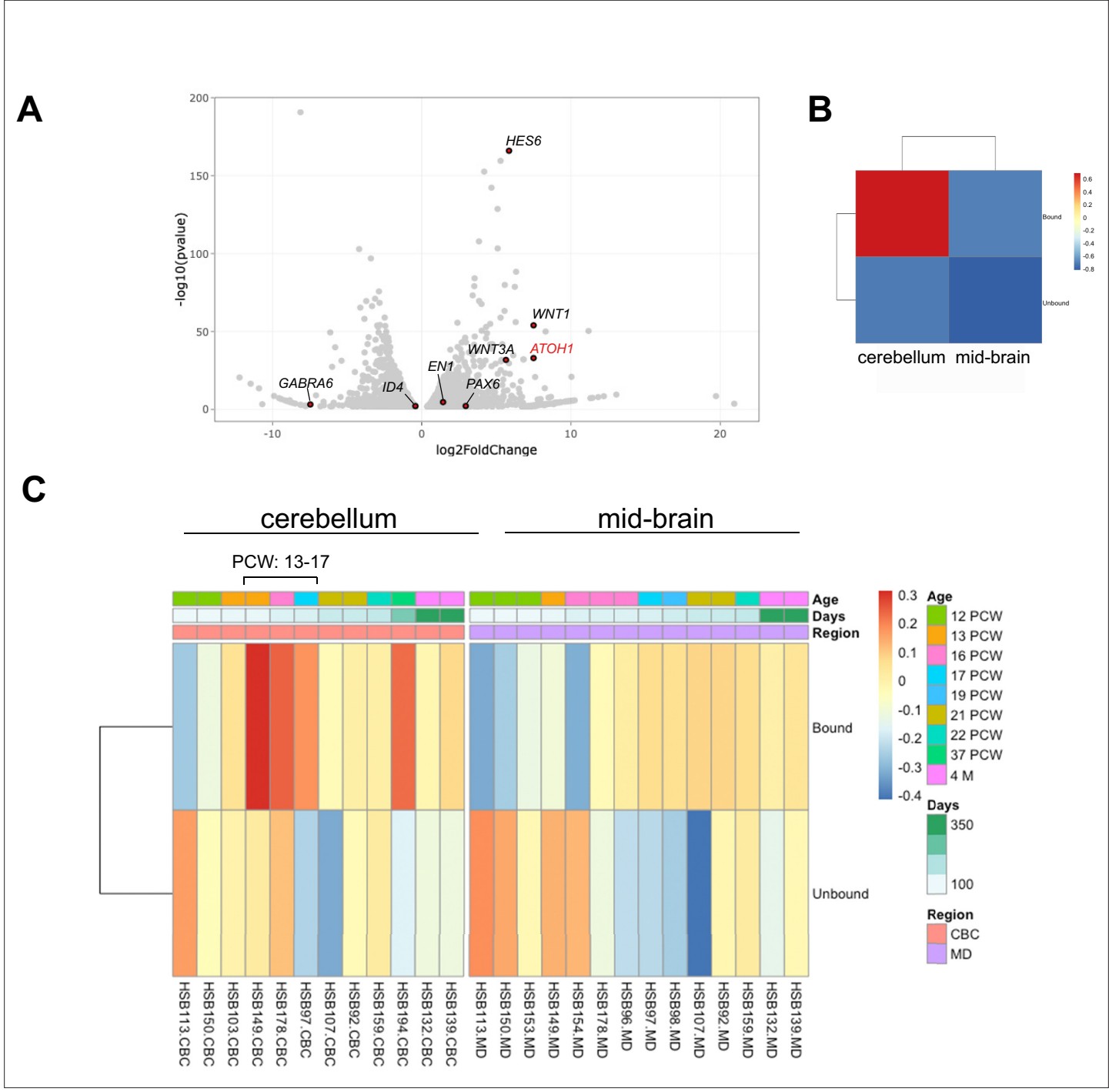

**Figure 4.** The human pluripotent stem cell (hPSC)-derived ATOH1 lineage resembles the human cerebellum in the second trimester by translational profiling. (**A**) Volcano plot of log$_2$ fold change global gene expression in ATOH1-TRAP IPs versus input. Key granule cell (GC) genes are highlighted by red dots (*Figure 4—source data 1*). The fully differentiated GC marker GABRA6 is depleted while progenitor genes are enriched. (**B**) Heatmap showing Gene Set Enrichment Analysis (GSEA) of log$_2$ fold-enriched genes in day in vitro (DIV) 28 ATOH1-TRAP versus the PsychEncode dataset for the developing human cerebellum and midbrain from 12 post coitus week (PCW) until 4 months of age (combined). (**C**) Heatmap of data in (**B**) but divided by timeline with columns representing our data (bound [IP] and unbound [input]) compared to individuals from the PsychEncode project (identifiers depicted at the bottom, *Figure 4—source data 2*). CBC, cerebellum; MB, midbrain.

The online version of this article includes the following source data and figure supplement(s) for figure 4:

**Source data 1.** DESeq2 analysis of *ATOH1-EGFP-L10a* TRAP IP versus input.

**Source data 2.** Comparison of *ATOH1-EGFP-L10a* TRAP IP to human developmental data from PsychEncode.

*Figure 4 continued on next page*

*Figure 4 continued*

**Figure supplement 1.** Heatmaps of key developmental signaling pathways in the hPSC-ATOH1 lineage.

**Figure supplement 2.** Comparison of *ATOH1-EGFP-L10a* TRAP against single-cell RNA-seq data from the developing mouse cerebellum.

**Figure supplement 2—source data 1.** Comparison of *ATOH1-EGFP-L10a* TRAP IP to scRNA-seq from *Wizeman et al., 2019*, See *Figure 4—figure supplement 2*.

development in the mouse (from E15.5 until P6) and multiple regions of the developing cerebellum revealed very few NEUROD1[+] cells at the pial surface of the EGL (*Figure 5—figure supplement 1B, C*). Only at P6, a relatively late stage of cerebellar development in the mouse, when GCP proliferation peaks (Ki67 throughout the EGL) and the EGL is at its thickest, did we detect an increase in NEUROD1[+] cells in the oEGL (*Figure 5—figure supplement 1B and C*). Punctate NeuN expression was detected in the human oEGL but not at any stages or regions examined in the mouse (*Figure 5D*, *Figure 5—figure supplement 1C*). Finally, since a large number of ATOH1-EGFP[+] cells were SOX2[+] in our cultures (*Figure 1—figure supplement 3C and D*) we examined SOX2 expression in the 17 PCW human cerebellum and found that SOX2 is extensively expressed in the human EGL (*Figure 5—figure supplement 1D*), in contrast to the mouse, where SOX2[+] cells in the EGL are a rare find (*Sutter et al., 2010*). Together, these data reveal marked species differences and extensive coexpression of transcriptional regulators that have classically until now been ascribed to either progenitors or differentiating neurons based on mouse studies.

## Discussion

We report a method for the scalable derivation of the human ATOH1 neuronal lineage from hPSCs that yields cerebellar progenitors by day 16 and GCs within 48 days in chemically defined medium. In contrast to previous methods for derivation of cerebellar neurons with a focus on Purkinje cells (*Buchholz et al., 2020*; *Erceg et al., 2010*; *Muguruma et al., 2015*; *Nayler et al., 2017*; *Silva et al., 2020*; *Wang et al., 2015*; *Watson et al., 2018*), we provide a timeline of developmental progression of the human ATOH1 lineage by gene expression and an unbiased comparison of the transcriptional profile to the developing human brain. While our focus was derivation of GCs, the expression of markers of the cerebellar nuclei (LHX9, LHX2) and the presence of Calretinin[+] cells indicate the production of additional ATOH1 derivatives. Future identification of cell surface markers for each class of glutamatergic neuron known to derive from the ATOH1 lineage will allow isolation and further characterization of the various populations yielded by our method. MAC-sorted early postmitotic GCs (TAG1[+]) integrated into the mouse cerebellum and migrated through the EGL into the IGL in close apposition to glial fibers, providing an in vivo system to study human GC migration and migration defects. This is the first successful demonstration of migration followed by integration of hPSC-GCs into a stratified cortex, suggesting that molecular pathways required for glial-guided migration have developed in the human cells.

Adaptation of TRAP-seq for use in hPSCs allowed lineage-specific molecular profiling of the ATOH1 lineage, providing a superior depth of read compared to scRNA-seq methods. Comparison of our data to RNA-seq data from the developing human brain matched the DIV28 hPSC-ATOH1 lineage to the human cerebellum at 13–17 PCW. Analysis of key developmental pathways revealed that the WNT pathway is particularly enriched. Indeed, mouse genetic studies show that WNT signaling is critical for both early cerebellar development and later circuit establishment (*Lucas and Salinas, 1997*; *McMahon and Bradley, 1990*).

Interestingly, using TRAP-seq methodology we identified a molecular divergence in the temporal expression of transcriptional regulators in progenitors in the human EGL, which in the mouse are expressed in cells that are further along the differentiation path. Genetic differences in developmental timing can cause birth defects or give rise to a novel morphology that could confer an evolutionary advantage. The divergent expression of NEUROD1 and RBFOX3 in the human oEGL suggests an expansion of an 'intermediate' cellular state that may serve to allow the progenitor pool to persist much longer, to enable the protracted period of cerebellar development in humans compared to mouse. Both NEUROD1 and RBFOX3 are expressed in early postmitotic as well as fully differentiated neurons, rarely in neuroblasts, across species and brain regions (*D'Amico et al., 2013*; *Lee et al., 1995*; *Miyata et al., 1999*; *Zhang et al., 2016*). Overexpression of either gene in progenitors induces

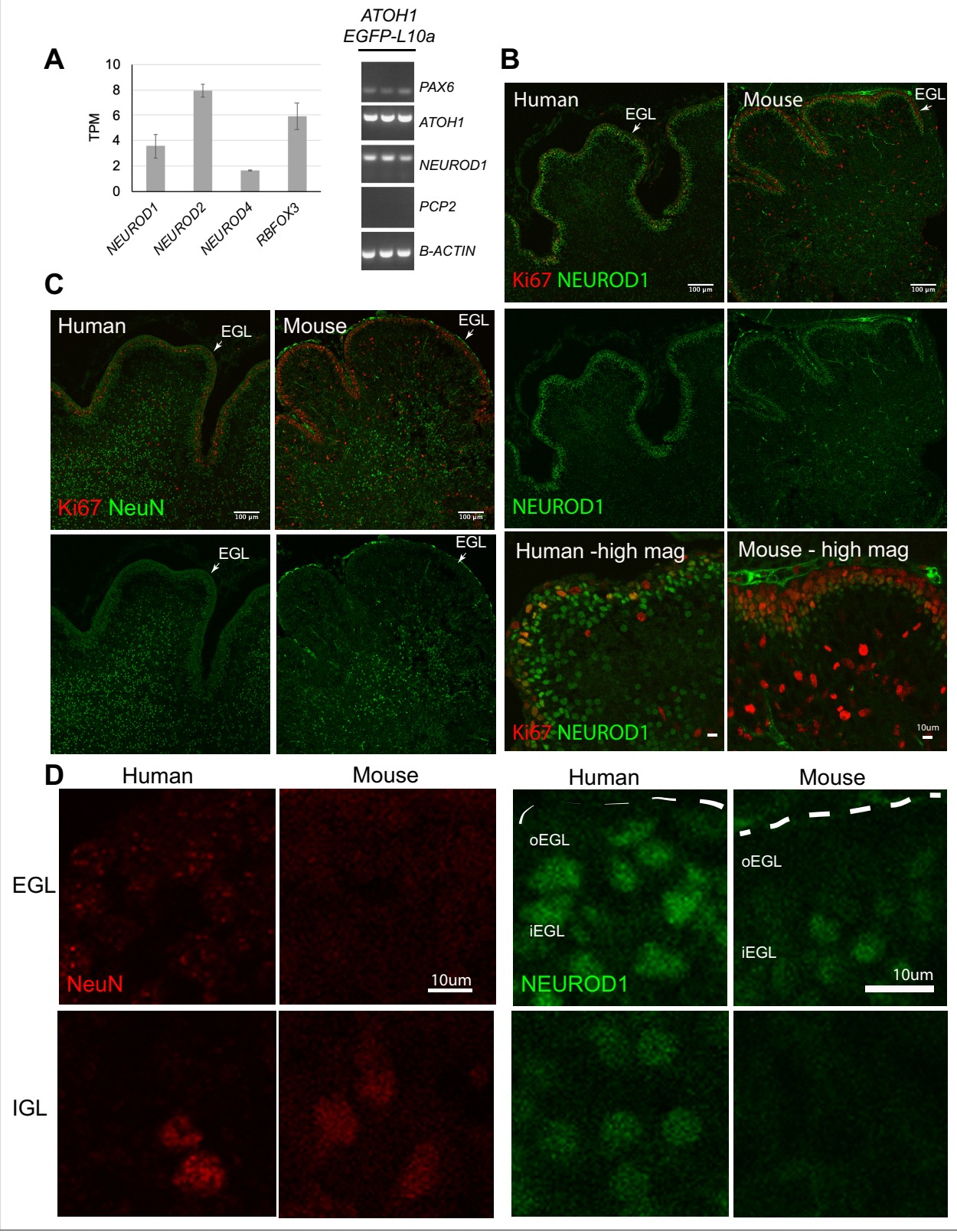

**Figure 5.** Temporal shift in the expression of transcriptional regulators in the human external granule cell layer (EGL) compared to mouse. (**A**) Left: bar chart showing the mean ± 1 SD normalized expression of transcriptional regulators in ATOH1-TRAP IPs at day in vitro (DIV) 28 by RNA-seq. Right: the expression of PAX6 (GCP marker), and coexpression of *ATOH1* and *NEUROD1*, but not *PCP2* (Purkinje cells marker) in ATOH1-TRAP IPs by RT-PCR. (**B**) Sagittal sections though the vermis showing NEUROD1 and Ki67 expression by immunohistochemistry in the human cerebellum at 17 post coitus week

*Figure 5 continued on next page*

*Figure 5 continued*
(PCW) and mouse at P0. Note the similarities in foliation depth and pattern. Bottom: higher magnifications (scale bars: 10 μm) of a lobule in human and mouse. (**C**) Mid-sagittal sections showing NeuN (RBFOX3) and Ki67 in the human (17 PCW) and mouse (P0). (**D**) Higher magnification of NeuN and NEUROD1 labeling in the human versus mouse EGL and internal granule cell layer (IGL) (scale bars: 10 μm). Left panel: note the punctate NeuN labeling in the human but not in the mouse EGL. Right panel: NEUROD1. Dashed lines demarcate the pial surface. N = 2 cerebella/species. o, outer; , inner; TPM, transcripts per million reads. Scale bars: 100 μm unless stated otherwise.

The online version of this article includes the following figure supplement(s) for figure 5:

**Figure supplement 1.** Marker expression in the developing human cerebellum versus mouse.

neuronal differentiation (*Boutin et al., 2010*; *Butts et al., 2014*; *Lee et al., 1995*; *Pataskar et al., 2016*; *Zhang et al., 2016*). Interestingly, in the *Xenopus* embryo, the coexpression of *Atoh1* and *NeuroD1* in the EGL has been reported, and it was hypothesized that both the timing of expression and the gene regulatory function of NEUROD1 have adapted to support development of a nonproliferative EGL (*Butts et al., 2014*). In the human, we propose that the coexpression of these factors in GCPs, together with the more extensive expression of SOX2 in the human EGL, may enable an extended immature (stem cell) or quiescent state to preserve the progenitor pool. We speculate that different levels of each factor, in combination, may be important for proliferation versus quiescent states in the GCPs. Indeed, Ki67 labeling showed fewer positive cells in the human EGL at 17 PCW than the comparable P0 mouse EGL, hinting at differences in proliferative regulation. Our findings, together with findings in the *Xenopus* and the differential proliferative capacity of the EGL in several vertebrate species (*Iulianella et al., 2019*), support the existence of evolutionary adaptations in EGL development across species. In future work, analyses of cell cycle length and comparison of molecular features in human versus mouse GCPs should provide further insights into the molecular basis of human cerebellar expansion and human-specific molecular mechanisms that will be critical for understanding medulloblastoma pathogenesis as well as cerebellar-mediated human cognitive evolution.

# Materials and methods

## Key resources table

| Reagent type (species) or resource | Designation | Source or reference | Identifiers | Additional information |
|---|---|---|---|---|
| Cell line (human) | RUES2 | Human embryonic stem cell line | NIH registration number: NIHhESC-09-0013 | |
| Cell line (human) | H9 (WA09) | Human embryonic stem cell line | NIH registration number: NIHhESC-10-0062 | |
| Cell line (human) | ATOH1-EGFP | Human embryonic stem cell line | | This study (RUES2 line) |
| Cell line (human) | ATOH1-EGFP-L10a | Human embryonic stem cell line | | This study (RUES2) |
| Recombinant DNA reagent | pPS-EF1α-GFP-RFP | System Biosciences | LV603PA-1 | Lentiviral vector |
| Recombinant DNA reagent | J2XnGFP | Dr. Jane Johnson | | GFP plasmid |
| Recombinant DNA reagent | pSIN-hATOH1 enhancer- hβ-globin-nlsEGFP-bGH polyA-hPGK-Puromycin | This study | | ATOH1-EGFP lentiviral construct |
| Recombinant DNA reagent | pSIN-hATOH1 enhancer- hβ-globin-nlsEGFP-bGH polyA-hPGK-Puromycin | This study | | ATOH1-EGFP-L10a lentiviral construct |
| Biological sample (human) | Human fetal cerebellum (17 PCW) | Human Developmental Biology Resource | | http://www.hdbr.org/ |
| Biological sample (mouse) | C57Bl/6J mice | Jackson Laboratory | | Embryos and pups obtained from times matings |
| Antibody | Calretinin (rabbit polyclonal) | Swant | 7699/4 | (1:1000) |
| Antibody | EN2 (C19) (goat polyclonal) | Santa Cruz | SC-8111 | (1:50) |
| Antibody | GFAP (chicken polyclonal) | EnCor | CPCA-GFAP | (1:1500) |

*Continued on next page*

*Continued*

| Reagent type (species) or resource | Designation | Source or reference | Identifiers | Additional information |
|---|---|---|---|---|
| Antibody | GFP (rabbit polyclonal) | Invitrogen | A-111122 | (1:500) |
| Antibody | GFP (chicken polyclonal) | Aves labs | GFP-1020 | (1:1000) |
| Antibody | HuNu (anti-human nuclei) (mouse monoclonal) | Millipore | MAB1281 | (1:100) |
| Antibody | Ki67 (rabbit monoclonal) | Vector Laboratories | VP-RM04 | (1:100) |
| Antibody | Ki67 (rabbit polyclonal) | EnCor | RPCA-Ki67 | (1:1000) |
| Antibody | Map2 (chicken polyclonal) | Abcam | ab5392 | (1:1000) |
| Antibody | NeuN (mouse monoclonal) | Millipore | MAB377 | (1:100) |
| Antibody | NeuroD1 (mouse monoclonal) | BD Pharmingen | 563000 | (1:300) |
| Antibody | Pax6 (rabbit polyclonal) | BioLegend | 901301 | (1:300) |
| Antibody | TAG1 (mouse monoclonal) | Tom Jessell | | (1:2) |
| Antibody | Synaptophysin (mouse monoclonal) | Millipore | MAB329 | (1:500) |
| Antibody | VGLuT1 (mouse monoclonal) | Millipore | MAB5502 | (1:100) |
| Antibody | SOX2 (rabbit monoclonal) | Cell Signaling | 3579 | (1:200) |
| Antibody | GluR-d2 (goat polyclonal) | Santa Cruz | Sc-26118 | (1:100) |
| Antibody | Anti-goat Alexa Fluor 633 (donkey polyclonal) | Invitrogen | A21082 | (1:300) |
| Antibody | Anti-rabbit Alexa Fluor 555 (donkey polyclonal) | Invitrogen | A-31572 | (1:300) |
| Antibody | Anti-mouse IgM Alexa Fluor 488 (goat polyclonal) | Invitrogen | A-21042 | (1:300) |
| Antibody | Anti-mouse Alexa Fluor 555 (donkey polyclonal) | Invitrogen | A-31570 | (1:300) |
| Antibody | Anti-chicken IgY 488 (donkey polyclonal) | Jackson ImmunoResearch | 703-545-155 | (1:300) |
| Antibody | Anti-chicken IgY Cy3 (donkey polyclonal) | Jackson ImmunoResearch | 703-165-155 | (1:300) |
| Antibody | Anti-rabbit Alexa Fluor 647 (donkey polyclonal) | Invitrogen | A-31573 | (1:300) |
| Antibody | Anti-mouse Alexa Fluor 647 (donkey polyclonal) | Invitrogen | A-31571 | (1:300) |
| Antibody | Anti-mouse Alexa Fluor 488 (donkey polyclonal) | Invitrogen | A-21202 | (1:300) |
| Antibody | Anti-mouse Fluor 405 (donkey polyclonal) | Abcam | Ab175658 | (1:300) |
| Sequence-based reagent | ATOH1 | This paper | PCR primer | Forward 5'-GCGCA AAAGAATTT GTCTCC-3' |
| Sequence-based reagent | ATOH1 | This paper | PCR primer | Reverse 5'-GCG AAGTTTTGCTG TTTTCC-3' |
| Sequence-based reagent | ID4 | This paper | PCR primer | Forward 5'-GC TCACTGCGCT CAACACC-3' |
| Sequence-based reagent | ID4 | This paper | PCR primer | Reverse 5'-GAA TGCTGTCGCC CTGCTTG-3' |
| Sequence-based reagent | EN2 | This paper | PCR primer | Forward 5'- GG CGTGGGTCTA CTGTACG-3' |

*Continued on next page*

*Continued*

| Reagent type (species) or resource | Designation | Source or reference | Identifiers | Additional information |
|---|---|---|---|---|
| Sequence-based reagent | EN2 | This paper | PCR primer | Reverse 5'-TACCTGTTG GTCTGGAA CTCG-3' |
| Sequence-based reagent | PAX6 | This paper | PCR primer | Forward 5'-TCA CCATGGCAA ATAACCTG-3' |
| Sequence-based reagent | PAX6 | This paper | PCR primer | Reverse 5'-CA GCATGCAGG AGTATGAGG-3' |
| Sequence-based reagent | NEUROD1 | This paper | PCR primer | Forward 5'-GGACGA GGAGCAC GAGGCAG ACAAGAA-3' |
| Sequence-based reagent | NEUROD1 | This paper | PCR primer | Reverse 5'-TTCCTCA GTGAGTCC TCCTCTG CGTTCA-3' |
| Sequence-based reagent | PCP2 | This paper | PCR primer | Forward 5'- GACC AGGAGGG CTTCTTCAATCT-3' |
| Sequence-based reagent | PCP2 | This paper | PCR primer | Reverse 5'- CATG TCCATGA GGCTGT CCATCT-3' |
| Sequence-based reagent | OTX2 | This paper | PCR primer | Forward 5'-ACAA GTGGC CAATTCA CTCC-3' |
| Sequence-based reagent | OTX2 | This paper | PCR primer | Reverse 5'-GAGG TGGACAA GGGATCTGA-3' |
| Sequence-based reagent | MEIS2 | This paper | PCR primer | Forward 5'-CCAG GGGACT ACGTTTCTCA-3' |
| Sequence-based reagent | MEIS2 | This paper | PCR primer | Reverse 5'-TAA CATTGT GGGGC TCTGTG-3' |
| Sequence-based reagent | GBX2 | This paper | PCR primer | Forward 5'-GTTCC CGCCG TCGCTGATGAT-3' |
| Sequence-based reagent | GBX2 | This paper | PCR primer | Reverse 5'-GCC GGTGTA GACGAA ATGGCCG-3' |
| Sequence-based reagent | HOXA2 | This paper | PCR primer | Forward 5-CGT CGCTC GCTGA GTGCCTG-3' |
| Sequence-based reagent | HOXA2 | This paper | PCR primer | Reverse 5'-TGTC GAGTGTG AAAGCG TCGAGG-3' |

*Continued on next page*

*Continued*

| Reagent type (species) or resource | Designation | Source or reference | Identifiers | Additional information |
|---|---|---|---|---|
| Sequence-based reagent | LHX2 | This paper | PCR primer | Forward 5'-GGTCCTC CAGGTCT GGTTC-3' |
| Sequence-based reagent | LHX2 | This paper | PCR primer | Reverse 5'-TAAGAG GTTGCGC CTGAACT-3' |
| Sequence-based reagent | LHX9 | This paper | PCR primer | Forward 5'- GCT GGGAGT GGACATCGTCA-3' |
| Sequence-based reagent | LHX9 | This paper | PCR primer | Reverse 5'- CATG GTCCGGA GCTGGTGAT-3' |
| Sequence-based reagent | *β-ACTIN* | This paper | PCR primer | Forward 5'-AAAC TGGAAC GGTGAAGG-3' |
| Sequence-based reagent | *β-ACTIN* | This paper | PCR primer | Reverse 5'-AGA GAAGT GGGGTGGCTT-3' |
| Sequence-based reagent | ATP5O | This paper | PCR primer | Forward 5'- cgcta tgccac agctcttta-3' |
| Sequence-based reagent | ATP5O | This paper | PCR primer | Reverse 5'- atgg aacgcttc acataggg-3' |
| Peptide, recombinant protein | Human bFGF | Invitrogen | Catalog # 13256-029 | |
| Peptide, recombinant protein | Human/mouse/ rat BDNF | PeproTech | Catalog # 450-02 | |
| Peptide, recombinant protein | Mouse BMP7 | R&D Systems | Catalog # 5666BP-010 | |
| Peptide, recombinant protein | Human recombinant insulin | Tocris | Catalog # 3435 | |
| Peptide, recombinant protein | Human BMP4 | R&D Systems | Catalog # 314BP-050 | |
| Peptide, recombinant protein | Human BMP6 | R&D Systems | Catalog # 507BP-020 | |
| Peptide, recombinant protein | Human/mouse FGF8b | R&D Systems | Catalog # 423-F8-025 | |
| Commercial assay or kit | RNeasy micro kit | QIAGEN | Catalog # 74004 | |
| Commercial assay or kit | RNeasy Plus mini kit | QIAGEN | Catalog # 74134 | |
| Commercial assay or kit | Transcription First Strand cDNA Synthesis Kit | Roche Life Sciences | Catalog # 04379012001 | |
| Commercial assay or kit | HotStarTaq PLUS DNA Polymerase kit | QIAGEN | Catalog # 203603 | |
| Commercial assay or kit | Click-iT EdU Cell Proliferation Kit for Imaging | Invitrogen | Catalog # C10338 | |
| Commercial assay or kit | SMART-Seq v4 Ultra Low Input RNA Kit | TaKaRa Bio | Catalog # 634888 | |
| Commercial assay or kit | Nextera XT DNA library preparation kit | Illumina | Catalog # FC-131-1024 | |
| Commercial assay or kit | RNA 6000 Pico Kit | Agilent | Catalog # 5067-1513 | |
| Commercial assay or kit | In-fusion HD cloning plus (Clontech) | TaKaRa Bio | Catalog # 638909 | |

*Continued on next page*

*Continued*

| Reagent type (species) or resource | Designation | Source or reference | Identifiers | Additional information |
|---|---|---|---|---|
| Commercial assay or kit | Anti-mouse IgM microbeads | Miltenyi Biotec | Catalog # 130-047-302 | |
| Commercial assay or kit | MS columns | Miltenyi Biotec | Catalog # 130-042-201 | |
| Chemical compound, drug | ROCK-inhibitor Y-27632 | Abcam | Catalog # ab 120129 | |
| Chemical compound, drug | SB431542 | Tocris | Catalog # 1614 | |
| Chemical compound, drug | LDN-193189 | Stemgent/ Tocris | Catalog # 6053 | |
| Chemical compound, drug | CHIR99021 | Stemgent/ REPROCELL | Catalog # 04-0004-02 | |
| Software, algorithm | Primer3 | Open source | | Primer3, RRID:SCR_003139 |
| Software, algorithm | Salmon quantification software (version 0.8.2) | Open source | | Salmon, RRID:SCR_017036 (*Patro et al., 2017*) |
| Software, algorithm | R statistical software | Open source | | R Project for Statistical Computing, RRID:SCR_001905 |
| Software, algorithm | Tximport (version 1.8.0). | Open source | | tximport, RRID:SCR_016752 (*Love et al., 2016*) |
| Software, algorithm | DESeq2 (version 1.20.0) | Open source | | DESeq2, RRID:SCR_015687 (*Love et al., 2018*) |
| Software, algoritham | | | | |
| Software, algorithm | rtracklayer package (version 1.40.6) | Open source | | rtracklayer, RRID:SCR_021325 |
| Software, algorithm | GSVA (version 1.34.0) | Open source (*Hänzelmann et al., 2013*) | | GSVA, RRID:SCR_021058 |
| Software, algorithm | Pheatmap R package (version 1.0.10) | Open source | ncv | pheatmap, RRID:SCR_016418 |
| Software, algorithm | topGO Bioconductor package | Open source | | topGO, RRID:SCR_014798 |
| Software, algorithm | GOseq Bioconductor package | Open source (*Young et al., 2010*) | | Goseq, RRID:SCR_017052 |
| Software, algorithm | ImageJ (version 2.1.0/1.53c) | Open source, NIH | | ImageJ, RRID:SCR_003070 |
| Software, algorithm | SPSS software | IBM | | |
| Software, algorithm | BD FACSDiva 8.0.1 software | BD Biosciences | | |
| Software, algorithm | ZEN imaging software | Zeiss | | |

## Human tissue collection, fixation, and embedding

Human prenatal brain tissue (two cerebella; 17 PCW) were acquired from the Human Developmental Biology Resource (http://www.hdbr.org/) following institutional policies. Tissues were fixed in 4% PFA for 7–10 days and washed multiple times in PBS. Samples were then cryoprotected in increasing concentrations of 5, 15, and 30% sucrose (in PBS) at 4°C. Samples were embedded in Tissue-Tek O.C.T. compound (VWR, 25608-930) and stored at –80°C until use.

## Mice

All procedures with mice were performed according to guidelines approved by the Rockefeller University Institutional Animal Care and Use Committee. C57Bl/6J mice (Jackson Laboratory) were maintained on a 12 hr light/dark cycle and a regular diet. Timed matings generated a mixture of male and female pups that were used for all described studies.

## hPSC culture

Human embryonic/pluripotent stem cells were used under approved institutional ESCRO committee protocols (The Rockefeller University). The RUES2 line (NIH #0013) was created, authenticated, and

provided by the lab of Dr. Ali Brivanlou at the Rockefeller University. The line was further karyotyped to confirm the sex (female) and karyotype (normal). The W9 line (NIH #0062) was obtained from Dr. Brivanlou's lab where it was routinely used in stem cell differentiation experiments along the three main embryonic lineages. hPSCs were maintained in growth media (HUESM medium conditioned with mouse embryonic fibroblasts and supplemented with 20 ng/ml bFGF [Invitrogen]) (*Deglincerti et al., 2016*). The growth medium was exchanged daily. For transgenic lines, Puromycin (Gibco) was added to the medium (1 µg/ml) during maintenance culture. Cells were grown as colonies on tissue culture dishes coated with hESC qualified Matrigel solution (Corning) in a 37°C humidified incubator with 5% $CO_2$.

## hPSC differentiation

hPSCs maintained as colonies were dissociated from plates with Trypsin-EDTA (0.25%, Gibco) for 4 min at 37°C in a humidified incubator. Cells were washed once with growth media (see previous section) and then resuspended in growth media with 10 µM ROCK-inhibitor Y-27632 (Abcam). Single cells were plated at 900 cells/ml on Transwell 6-well plates with permeable 24 mm polyester membrane inserts (Corning, 3450, matrigel-coated) or regular tissue culture-treated plates (matrigel-coated). On Transwell dishes, cells were plated on top of the membrane with 1 ml growth medium plus ROCK-inhibitor added below and above the membrane (2 ml total). The next day, 1 ml growth medium plus ROCK-inhibitor was added to the top part of the Transwell (3 ml total). On day 2, the medium was switched to differentiation medium: DMEM/F12 with sodium bicarbonate (Invitrogen, 11320-033), 0.5% BSA, 0.1 mM β-mercaptoethanol, 2 mM glutamate, 10 µM NEAA, 1× N2 supplement, 1× B27 without retinoic acid (all from Gibco) and Primocin (0.1 mg/ml, InvivoGen). This marked day 0 of differentiation. For differentiation experiments with transgenic lines, Puromycin (Gibco) was added to the medium (0.5 µg/ml) until DIV28. The following small molecules and growth factors were added to the differentiation medium on days indicated in *Figure 1D* as follows: 10 µM SB431542 (SB, Tocris) and 100 nM LDN-193189 (Stemgent) at days 0–7, 2.5 µM CHIR99021 (Stemgent) at days 1–11, 20 ng/ml bFGF (Invitrogen) at days 1–28, 25 ng/ml BDNF at days 11–28, and 50 ng/ml recombinant mouse BMP7 (R&D Systems) at days 7–15. Medium was exchanged for fresh differentiation medium plus appropriate factors every other day until day 28. The following conditions were also tested (not all data are included in the article) to arrive to the optimized protocol for the derivation of the ATOH1 lineage: CHIR99021 (Stemgent) at days 1–11 (range tested: 1–3 µM), 7 µg/ml human recombinant Insulin (Tocris), 100 ng/ml recombinant human/mouse FGF8b (Tocris) between days 1–11 (range tested: 50–500 ng/ml), recombinant mouse BMP7 (R&D Systems) at days 7–15 (range tested: 20–1000 ng/ml), 20 ng/ml BMP6 (R&D Systems), BMP4 (range tested: 4–160 ng/ml, R&D Systems), and 100 ng/ml GDF7 (R&D Systems). Depending on culture conditions used, such as culture surface, the concentration of CHIR99021 may need to be adjusted to achieve optimal anterior hindbrain patterning.

## Generation of transgenic hESC lines

For derivation of the *ATOH1-EGFP* line, a human *ATOH1* enhancer sequence (GenBank accession number AF218259.1; *Helms et al., 2000*) was amplified from human genomic DNA using two in-fusion primers (forward primer: 5′-TTCAAAATTTTATCGATaaggttcttCTATGGAGTTTGCA-3′; reverse primer: 5′- AATAGGGCCCTCTAGAGAATTCCTGAACAACCCCAC-3′). The amplicon was cloned into a modified version of the self-inactivating lentiviral vector pPS-EF1α-GFP-RFP (System Biosciences, LV603PA-1) where GFP and RFP had been removed and replaced by an hPGK-Puromycin cassette (pSIN-EF1a promoter-BGH polyA-hPGK-Puromycin). The vector was digested with *ClaI* and *XbaI* enzymes to remove the *EF1a* promoter, and the *ATOH1* enhancer sequence was cloned using in-fusion HD cloning (Clontech). Subsequently, a sequence containing the human beta globin minimal promoter (hβ–*globin*) followed by a nuclear localization signal and *EGFP* (nls-EGFP) was amplified from the J2XnGFP plasmid DNA (a gift from Dr. Jane Johnson, UT Southwestern) and cloned downstream of the human *ATOH1* enhancer by in-fusion HD cloning with the following primers (nls-EFGFP forward primer: 5′-GGTTGTTCAGGAATTCGATGGGCTGGGCATAAAAGT-3′; nls-EFGFP reverse primer: 5′- GCCCTCTAGAGAATTCAACTAGAGGCACAGTCGAGGC-3′) to obtain the following lentiviral construct: pSIN-hATOH1 enhancer-hβ-globin-nlsEGFP-bGH polyA-hPGK-Puromycin. For derivation of the *ATOH1-EGFP-L10a* line, the *nlsEGFP* was replaced with *EGFP-L10a*. Briefly, the

pSIN-hATOH1 enhancer- hβ-globin-nlsEGFP-bGH polyA-hPGK-Puromycin construct was digested with *SbfI/BsrGI* enzymes to remove the *nlsEGFP*. An *EGFP-L10a* fusion fragment was amplified from a template plasmid (mPCP2-A box-s296, a gift from Dr. Nathaniel Heintz, The Rockefeller University) with the following in-fusion PCR primers (EGFP-L10a-forward primer: 5′- CATTTGCTTCTAGCCT GCAGGTCGCCACCATGGTGAG-3′; and EGFP-L10a reverse primer: 5′- CCGCTTTACTTGTACATTAT CTAGATCCGGTGGATCC-3′) and cloned by in-fusion HD cloning to obtain pSIN-hATOH1 enhancer-EGFP-L10a-bGH polyA-hPGK-Puromycin. Lentiviral-mediated delivery was used according to established protocols to introduce the vectors into the RUES2 hESC line to generate transgenic lines. Two clonal lines with normal karyotype (Molecular Cytogenetics Core facility, MSKCC) and pluripotency characteristics were selected and expanded per transgenic line. Upon differentiation, EGFP expression was examined by microscopy. In the *ATOH1-EGFP* line, EGFP was broadly expressed in the nuclei of a subset of cells, while in the *ATOH1-EGFP-L10a* line, EGFP puncta were localized to the nucleoli, the site of ribosomal biogenesis (*Figure 1—figure supplement 2A and B*). A majority of the labeled cells coexpressed Ki67 (proliferation marker, data not shown), and the *ATOH1* transcript was enriched in FAC-sorted EGFP+ cells from the *ATOH1-EGFP* line compared to EGFP- cells, and upon IP in the *ATOH1-EGFP-L10a* line compared to input. By contrast, housekeeping genes were not enriched (*Figure 1—figure supplement 2A and B*).

## Fluorescence-activated cell sorting (FACS)

Cells at days 16, 19, 23, and 28 of differentiation were FAC-sorted on a BD FACSAriaII with BD FACS-Diva 8.0.1 software (BD Biosciences) using a 100 µm nozzle and a 488 nm laser according to standard procedure. Briefly, differentiation cultures were washed once in $Ca^{2+}/Mg^{2+}$-free PBS and the cells were dissociated by incubation in Accutase (Millipore, SCR005) for 5 min at 37°C. Dissociated cells were resuspended in MACS buffer (see MACS sections) and put through a cell strainer (BD Falcon, 352235). Gating was performed on EGFP-positive and -negative control cells, and propidium iodide (Sigma), at an appropriate dilution, was used for dead cell exclusion. EGFP+ cells were collected in MACS buffer for further downstream analyses.

## MACS of TAG1+ cells

On days 28–32 of differentiation, TAG1+ cells were isolated by MACS (Miltenyi Biotec) according to the manufacturer's protocol. Briefly, cells were washed 1× in MACS buffer (0.5% BSA, 0.9% glucose in $Ca^{2+}/Mg^{2+}$-free PBS) and then gently scraped off from Transwell membranes in MACS buffer using a cell scraper (USA Scientific, CC7600-0220). Cells were collected by centrifugation (300 × *g*, 10 min). 1 ml Trypsin (1 g/ml)-DNase (100 mg/ml) solution (Worthington Biochemical, 3703 and 2139) was added to the pellet for 1.5 min at 37°C without disturbing the pellet. The Trypsin-DNase was then exchanged for 1 ml DNase (100 mg/ml), and the cell pellet was triturated using a fine-bore pulled glass pipette until a uniform cell suspension was obtained. The suspension was put through a 40 µm cell strainer (BD Biosciences, 352340) to remove remaining clumps. Serum containing 'cerebellum' media (see subsequent section, plus 10% horse serum, Invitrogen 26050-088) was added to inactivate the Trypsin-DNase, and the single-cell suspension was washed and spun (300 × *g*, 10 min) in 50 ml of CMF-PBS (PBS Thermo Fisher, 14190-250; 0.2% w/v glucose Millipore, G8769; 0.004% v/v $NAHCO_3$ Millipore, S8761; 0.00025% Phenol Red Millipore, P0290). The cells were resuspended in 1 ml serum containing 'cerebellum' medium and spun in a tabletop centrifuge (300 × *g*, 5 min) in an Eppendorf tube. This step helped reduce dead cells and debris, which accumulated in the supernatant. The cell pellet was then resuspended in fresh 'cerebellum' medium plus serum and incubated in a bacterial dish in a 35°C, 5% $CO_2$ incubator for 1 hr. This step allowed surface antigens to reappear after the enzymatic dissociation of cells. The number of live cells was counted, and the cells were incubated in TAG1 antibody in cerebellum medium (1:2) for 20 min at room temperature (RT), followed by 1× wash in MACS buffer, and then incubated in anti-mouse IgM microbeads (Miltenyi Biotec, 130-047-302) for 15 min at RT, and TAG1+ cells were sorted through MS columns (Miltenyi Biotec, 130-042-201) following the manufacturer's description.

## Purification of cerebellar neurons and glia and co-culture with human cells

Mixed cerebellar cultures or glial fractions were prepared from P0-1 pups and cultured in serum-free 'cerebellum' media (BME, Gibco; 2 mM L-glutamate, Gibco; 1% v/v BSA, Sigma; ITS liquid media supplement, Sigma; 0.9% v/v glucose, Sigma; 0.1 mg/ml Primocin, InvivoGen) as previously described (*Baptista et al., 1994*; *Hatten, 1985*). Mixed cerebellar cultures (no separation step) were plated at $2.8 \times 10^6$ cells/ml on poly-D-lysine (Millipore)-coated glass coverslips (Fisher, 12-545-81) placed in 24-well plates. Glia-only fractions (separated by Percoll gradient) were plated at $0.8 \times 10^6$ cells/ml. TAG1+ human cells isolated at DIV28 by MACS were then plated on top of mixed cerebellar cultures the next day (at $0.3–0.5 \times 10^6$ cells/ml). Glial cultures were allowed to form a monolayer (5–7 days) upon which DIV28 TAG1+ cells were then plated. As previously reported for their mouse counterparts (*Hatten, 1985*), the ratio of glia to human neurons in co-culture was crucial. At 1:4 (glia:neuron), the neurons induced detachment of glia from the culture dish and co-aggregated into attached spheres while at a 1:2 (glia:neuron) ratio the glia remained as a bed upon which neurons attached and extended processes. Half of the medium (serum-free cerebellum media) was replaced with fresh medium every 4 days. For the in vitro migration assays, glia were isolated from P0-1 pups as described above and plated on poly-D-lysine-coated glass coverslips. The next day, DIV28 TAG1-sorted human cells were plated on top in a 1:2 ratio (glia:neuron) in the above described medium. Cells were fixed at 36 hr in 4% PFA for 15 min at RT and processed for immunolabeling.

## Gene expression analysis by RT-PCR and RT-qPCR

mRNAs were extracted using the RNeasy Plus mini kit (QIAGEN) with on column genomic DNA elimination, and cDNAs transcribed with the Transcription First Strand cDNA Synthesis Kit (Roche) according to the manufacturer's description. Reverse transcription and PCR were carried out according to the manufacturer's descriptions using the HotStarTaq PLUS DNA Polymerase kit (QIAGEN) on a PTC-200 Peltier Thermal Cycler (MS Research). To catch cDNA amplification in the exponential phase, experiments were run for 30 cycles only as follows: initial heat activation at 95°C for 5 min, denaturation at 94°C for 30 s, annealing at primer-specific temperatures listed in *Table 1* for 40 s, extension at 72°C for 1 min, final extension at 72°C for 7 min. RT-qPCR was performed using the SYBR Green method according to the manufacturers' descriptions (Roche) using the default SYBR Green program on a Roche LightCycler 480 (Roche). All experiments were performed on at least three independent biological replicates, and each sample was run in triplicate for RT-qPCR. Data were normalized to

**Table 1.** Table of primers.

| Gene | Forward primer | Reverse primer | °C |
|------|----------------|----------------|-----|
| ATOH1 | 5'-GCGCAAAAGAATTTGTCTCC-3' | 5'-GCGAAGTTTTGCTGTTTTCC-3' | 60 |
| ID4 | 5'-GCTCACTGCGCTCAACACC-3' | 5'-GAATGCTGTCGCCCTGCTTG-3' | 60 |
| EN2 | 5'-GGCGTGGGTCTACTGTACG-3' | 5'-TACCTGTTGGTCTGGAACTCG-3' | 59 |
| PAX6 | 5'-TCACCATGGCAAATAACCTG-3' | 5'-CAGCATGCAGGAGTATGAGG-3' | 60 |
| NEUROD1 | 5'-GGACGAGGAGCACGAGGCAGACAAGAA-3' | 5'-TTCCTCAGTGAGTCCTCCTCTGCGTTCA-3' | 56 |
| PCP2 | 5'-GACCAGGAGGGCTTCTTCAATCT –3' | 5'-CATGTCCATGAGGCTGTCCATCT-3' | 56 |
| OTX2 | 5'-ACAAGTGGCCAATTCACTCC-3' | 5'-GAGGTGGACAAGGGATCTGA-3' | 60 |
| MEIS2 | 5'-CCAGGGGACTACGTTTCTCA-3' | 5'-TAACATTGTGGGGCTCTGTG-3' | 50 |
| GBX2 | 5'-GTTCCCGCCGTCGCTGATGAT-3' | 5'-GCCGGTGTAGACGAAATGGCCG-3' | 60 |
| HOXA2 | 5-CGTCGCTCGCTGAGTGCCTG-3' | 5'-TGTCGAGTGTGAAAGCGTCGAGG-3' | 60 |
| LHX2 | 5'- GGTCCTCCAGGTCTGGTTC-3' | 5'-TAAGAGGTTGCGCCTGAACT-3' | 60 |
| LHX9 | 5'- GCTGGGAGTGGACATCGTCA-3' | 5'-CATGGTCCGGAGCTGGTGAT-3' | 60 |
| β-ACTIN | 5'-AAACTGGAACGGTGAAGG-3' | 5'-AGAGAAGTGGGGTGGCTT-3' | 59 |
| ATP5O | 5'-cgctatgccacagctttta-3' | 5'-atggaacgcttcacataggg-3' | 60 |

housekeeping genes for comparisons and fold change calculated by the $2^{\Delta\Delta CT}$ method. *Figure 1B* and *Figure 1—figure supplement 2A* show technical replicates. As *ATOH1* was not detected by the end of the RT-qPCR at cycle number 45 (but the housekeeping gene was) in the EGFP- samples, 45 was assigned as the *ATOH1* cycle number value (CT) to signify a nondetected signal and calculate the fold change in the EGFP+ samples compared to EGFP- samples. For primer sequences and annealing temperatures, see *Table 1*. Primers were designed using Primer3 (https://bioinfo.ut.ee/primer3-0.4.0/) and validated to give single amplicons in a concentration-dependent manner at temperatures used.

## Immunocyto/histochemistry

Cells grown on 1.5 thickness cover glass (Fisher) were fixed for 15 min at RT with 4% PFA and washed 3× PBS. Cells were then blocked in PBS containing 1% normal horse serum (Gibco) and 0.1% Triton for 1 hr and then incubated with primary antibodies in blocking solution for 1 hr at RT to overnight at 4°C. Cells were washed 3× PBS for 10 min, and Alexa Fluor conjugated secondary antibody incubations were performed in blocking solution for 1 hr at RT. Dapi was sometimes added as nuclear counterstain

**Table 2.** Table of antibodies.

| Antibody | Species | Source | Dilution | Catalog # |
|---|---|---|---|---|
| Calretinin | Rabbit | Swant | 1:1000 | 7699/4 |
| EN2 (C19) | Goat | Santa Cruz | 1:50 | SC-8111 |
| GFAP | Chicken | EnCor Biotechnology | 1:1500 | CPCA-GFAP |
| GFP | Rabbit | Invitrogen | 1:500 | A-111122 |
| GFP | Chicken | Aves labs | 1:1000 | GFP-1020 |
| HuNu (anti human nuclei) | Mouse | Millipore | 1:100 | MAB1281 |
| Ki67 | Rabbit | Vector Laboratories | 1:100 | VP-RM04 |
| Ki67 | Rabbit | EnCor | 1:1000 | RPCA-Ki67 |
| MAP2 | Chicken | Abcam | 1:1000 | ab5392 |
| NEUN | Mouse | Millipore | 1:100 | MAB377 |
| NEUROD1 | Mouse | BD Pharmingen | 1:300 | 563000 |
| PAX6 | Rabbit | BioLegend | 1:300 | 901301 |
| TAG1 | Mouse IgM | T.Jessell | 1:2 | N/A |
| Synaptophysin | Mouse IgM | Millipore | 1:500 | MAB329 |
| VGLUT1 | Mouse | Millipore | 1:100 | MAB5502 |
| SOX2 (D6D9) | Rabbit | Cell Signaling | 1:200 | 3579 |
| GluR-δ2 | Goat | Santa Cruz | 1:100 | Sc-26118 |
| Anti-goat Alexa Fluor 633 | Donkey | Invitrogen | 1:300 | A21082 |
| Anti-rabbit Alexa Fluor 555 | Donkey | Invitrogen | 1:300 | A-31572 |
| Anti-mouse IgM Alexa Fluor 488 | Goat | Invitrogen | 1:300 | A-21042 |
| Anti-mouse Alexa Fluor 555 | Donkey | Invitrogen | 1:300 | A-31570 |
| Anti-chicken IgY 488 | Donkey | Jackson ImmunoResearch | 1:300 | 703-545-155 |
| Anti-chicken IgY Cy3 | | Jackson ImmunoResearch | 1:300 | 703-165-155 |
| Anti-rabbit Alexa Fluor 647 | Donkey | Invitrogen | 1:300 | A-31573 |
| Anti-mouse 405 | Donkey | Abcam | 1:300 | Ab175658 |
| Anti-mouse Alexa Fluor 647 | Donkey | Invitrogen | 1:300 | A-31571 |
| Anti-mouse Alexa Fluor 488 | Donkey | Invitrogen | 1:300 | A-21202 |

(1 µg/ml, Molecular Probes). For immunohistochemistry on frozen brain sections, human cerebella were fixed and embedded as described earlier. Mouse brains at E15.5, E17.5, P0, and P6 were fixed in 4% PFA overnight at 4°C, then cryoprotected in 20% sucrose overnight at 4°C and embedded in OCT. 14-µm-thick sagittal sections were prepared for all brains on a Leica CM 3050S cryostat. Frozen sections were thawed, postfixed for 10 min in 4% PFA at RT, and immunohistochemistry was carried out as above, except that the blocking solution contained 10% normal horse serum (Gibco) and 0.2% Triton in PBS. For analysis of transplantation experiments, brains were fixed in 4% PFA overnight at 4°C, washed in PBS, and embedded in 3% agarose. 50-µm-thick vibratome sections were postfixed with 4% PFA for 15 min at RT followed by blocking in 10% normal horse serum (Gibco) and 0.2% Triton in PBS overnight at 4°C. Primary antibody incubations were carried out in blocking solution for two nights at 4°C followed by extensive washes (4 × 15 min each) in PBS containing 0.1% Triton, and sections were then incubated in secondary antibodies overnight at 4°C. Sections/cells were mounted with ProLong Gold anti-fade mounting media (Invitrogen) and 1.5 thickness Fisherbrand cover glass. For antibody sources and dilutions, see *Table 2*.

## Proliferation assay

Cell proliferation was measured by EdU incorporation using the Click-iT EdU Cell Proliferation Kit for Imaging (Invitrogen, C10338) according to the manufacturer's description. On day 11 of differentiation, cells from the *ATOH1-EGFP* line were gently scraped off Transwell membranes and plated on poly-D-lysine (Millipore, A-003-E) and Laminin (Invitrogen, 23017-015)-coated glass coverslips, and differentiation was resumed as described previously (*Figure 1D*). On day 28, EdU was added to the culture medium according to the manufacturer's description and cells were treated with either SAG (0.5 µM, Cayman Chemicals 11914) or DMSO. Cells were fixed with 4% PFA (15 min at RT) on day 30 and processed for Click-iT EdU detection and immunocytochemistry. See below for quantification and statistical analysis.

## Transplantation of hPSC-derived cerebellar granule cells in the mouse cerebellum

All procedures were approved by the Rockefeller University Institutional Animal Care and Use Committee. TAG1$^+$ cells were isolated by MACS on DIV28–32 as described earlier. Cells were counted and resuspended in 'transplantation medium' containing: BME, Gibco; 10% v/v horse serum, Invitrogen 26050-088; 0.9% v/v glucose, Sigma; and 0.5% w/v Fast Green for visualization of injection volumes. Cells were kept on ice while mouse pups were prepared for injections (up to 2 hr). Neonatal mouse pups (P1-4) were cryo-anesthetized, the heads were cleaned with alcohol, and a 1 µl single-cell suspension (5 × 10$^6$ cells/ml) was manually injected directly into the left cerebellar hemisphere using a glass microcapillary (Eppendorf, 5195 000.079) controlled with an Eppendorf CellTram Vario manual microinjector. The procedure was performed under a Zeiss OMPI-1FC surgical microscope (ENT) with Zeiss eyepieces. The capillary directly pierced through both skin and skull. The positioning of the capillary was guided by the left earlobe and Lambda, and the tip of the capillary was placed just under the skull to target the cerebellar surface. The capillary was held in place for 1 min after the completion of injection to minimize backflow after which it was gently pulled out and the pups were warmed up on a heating pad (Sunbeam) before being returned to their mothers. Injected animals were analyzed 48 hr post injection. Brains were dissected out and fixed overnight in 4% PFA. The entire cerebellum of each animal (N = 5 analyzed) was sectioned coronally at 50 µm thickness on a vibratome (Leica VT 1000S), and all sections were processed for immunohistochemistry using a human nuclear antigen antibody (HuNu) to detect human cells plus additional antibodies as described in the article.

## TRAP and RNA sequencing

TRAP was performed as previously reported (*Heiman et al., 2014*) on three independent differentiation experiments at DIV28. Briefly, polysomes were stabilized by adding 100 µg/ml cycloheximide to cell culture media for 10 min prior to homogenization of cells with polysome extraction buffer. Following clearing by centrifugation, supernatants were incubated at 4°C with end-over-end rotation for 16–18 hr with biotinylated Streptavidin T1 Dynabeads (Thermo Fisher, 65601) previously conjugated with GFP antibodies (Sloan Kettering Institute Antibody Core, HtzGFP-19C8 and HtzG-FP-19F7). The beads were collected on a magnetic rack, washed, and resuspended in lysis buffer with

β-mercaptoethanol (Agilent, 400753) to extract bound RNA from polysomes. RNA was purified using the RNeasy micro kit (QIAGEN, 74004). RNA quantity and quality were measured using an Agilent 2100 Bioanalyzer with the 6000 Pico Kit (Agilent, 5067-1513). Full-length cDNA was prepared using Clontech's SMART-Seq v4 Ultra Low Input RNA Kit (634888) from 0.5 ng RNA with an RIN ≥ 8.3. 1 ng cDNA was then used to prepare libraries using the Illumina Nextera XT DNA sample preparation kit (FC-131-1024). Libraries with unique barcodes were pooled at equal molar ratios and sequenced on Illumina NextSeq 500 sequencer to generate 75 bp single reads, following the manufacturer's protocol.

## RNA sequencing analysis

Sequence and transcript coordinates for human hg19 UCSC genome and gene models were retrieved from the Bioconductor Bsgenome.Hsapiens.UCSC.hg19 (version 1.4.0) and TxDb.Hsapiens.UCSC.hg19.knownGene (version 3.2.2) Bioconductor libraries, respectively. Transcript expressions were calculated using the Salmon quantification software (*Patro et al., 2017*) (version 0.8.2) from raw FastQ files. Gene expression levels as TPMs and counts were retrieved using Tximport (*Love et al., 2016*) (version 1.8.0). Normalization and rlog transformation of raw read counts in genes were performed using DESeq2 (*Love et al., 2018*) (version 1.20.0). For visualization in genome browsers, RNA-seq reads were aligned to the genome using Rsubread's subjunc method (version 1.30.6) (*Liao et al., 2013*) and exported as bigWigs normalized to reads per million using the rtracklayer package (version 1.40.6). Genes significantly enriched or depleted in IP over input were identified using DESeq2 with a Benjamini–Hochberg adjusted p-value cutoff of 0.05 and absolute log fold change cut-offs of both 0 and 2. The PsychEncode's 'Human mRNA-seq processed data' as counts was retrieved from the PsychEncode's portal (http://development.psychencode.org). GSEA of significantly enriched or depleted gene sets (absolute logFC > 2, padj<0.05) were performed using GSVA (version 1.34.0) (*Hänzelmann et al., 2013*) against DESeq2 normalized PsychEncode midbrain and cerebellum RNA-seq counts. Statistical significance of gene set enrichment within samples was determined using Limma's geneSetTest with normalized median scaled expression values. Visualization of genes and gene sets as heatmaps was performed using the Pheatmap R package (version 1.0.10) (*Subramanian et al., 2005*). GO term enrichment was obtained for all genes differentially expressed between IP and input (absolute logFC > 0, adjusted p-value<0.05) using the Fisher test in the topGO Bioconductor package and ranked using the elim algorithm and functional annotation from the org.Hs.eg.db Bioconductor package (version 3.10). For comparison to mouse scRNA-seq data, cell-type marker gene sets were retrieved from Wizeman et al. and mouse symbols translated to human ortholog symbols. Genes significantly upregulated in IP over input (logFC > 0, adjusted p-value<0.05) were tested for enrichment of cell type markers using the GOseq Bioconductor package (*Young et al., 2010*).

## Imaging

Single z-plane images (512 × 512 pixels) were acquired using an inverted Zeiss LSM 880 NLO laser scanning confocal microscope operated with ZEN imaging software (Zeiss) and fitted with a Plan-Apochromat 40×/1.4 NA objective oil immersion lens, a Nomarski prism, and HeNe and Argon lasers for excitation at 405, 488, 561, and 633 nm. Phase-contrast images in *Figure 1—figure supplement 1D* were acquired with a 20× objective on a Leica DMIL LED microscope fitted with a Leica MC120 HD camera.

## Quantification and statistical analyses

Sample size estimations were based on prior pilot experiments. Independent experiments, performed on different days (cell culture experiments) or independent samples (e.g., individual mice), were considered biological replicates. Repeat measurements on the same sample were considered technical replicates. Samples were randomly allocated to control and treatment groups. For immunocyto-/histochemistry experiments, cells were manually counted in ImageJ (version 2.1.0/1.53c) on single z-plane confocal images (512 × 512 pixels) from three independent (in vitro) experiments (unless stated otherwise). Data were checked for normal distribution by Shapiro–Wilk's test of normality and analyzed by parametric tests as described below using SPSS software (IBM). In *Figure 1E*, the mean percentages of ATOH1-EGFP+ cells were compared by ANOVA. The percentages of PAX6, EN2, and EGFP-positive cells were calculated per PAX6, EN2, and EGFP populations as indicated in *Figure 1F*.

Multiple representative images per coverslip were quantified, and a total of 1787 EN2$^+$, 3487 GFP$^+$, and 2845 PAX6$^+$ cells were counted. For SAG treatment experiments, the percentage of EdU/EGFP double-positive cells was calculated as a subset of the Dapi population. A total of 84 images (6209 EdU$^+$ cells) were analyzed with five outliers removed. Outliers were defined as datapoints that were 1.5× outside of the interquartile range of box plots. Data were log transformed for normality, and the SAG and control treatment groups were compared by ANCOVA with the number of Dapi$^+$ cells as a covariate. The number of TAG1$^+$ cells was counted after MACS using a hemocytometer under a Leica microscope (Leica DMIL LED) and expressed as a percentage of the cells at the start of the sort (input). All other markers (NEUROD1, NeuN, Calretinin, PAX6, SOX2) were quantified on multiple confocal z-plane images in ImageJ from three independent experiments (except SOX2, n = 2) as described earlier. For the Tag1$^+$ fraction, the number of cells quantified for each marker is indicated in the main text. For the SOX2 quantifications, a total of 2376 cells were counted. For the TAG1$^-$ fraction (*Figure 2—figure supplement 1*), a total of 2714 cells were counted. The percentage of NEUROD1-positive cells at the pial surface of the cerebellum was counted and normalized to the number of Dapi$^+$ nuclei along the length of the pial surface. Multiple sagittal sections from two brains per time point and species were analyzed with the total numbers of cells/sample indicated in the description of *Figure 5—figure supplement 1*.

## Acknowledgements

We thank Dr. Ali Brivanlou and Dr. Zeeshan Ozair (The Rockefeller University) for the provision of vital reagents and critical discussions throughout the project, Dr. Nathaniel Heintz and Jie Xing (The Rockefeller University) for the kind gift of the EGFP-L10a containing construct, helpful discussions regarding the adaptation of bacTRAP methodology for use in hPSCs and technical support. We also thank Dr. Bobak Mosadegh (Cornell University) for discussions on the use of transwells and minimization of culture variability, Dr. Jane Johnson (UT Southwestern) for the kind gift of the J2XnGFP plasmid, and the late Dr. Tom Jessell (Columbia University) for the provision of the TAG1 antibody. We are also grateful to staff at the Rockefeller Core facilities including imaging, flow cytometry, and high-throughput sequencing. Finally, we thank members of the Hatten lab for helpful discussions. This work was supported by NIH 1R21NS093540-01 (MEH), a Rockefeller University Center for Clinical and Translational Science Pilot grant (MEH, HB), a Tri-Institutional Stem Cell Initiative grant from the Starr Foundation (MEH), and a gift from the Renate Hans and Maria Hofmann Trust (MEH).

## Additional information

### Competing interests

Mary E Hatten: Reviewing editor, eLife. The other authors declare that no competing interests exist.

### Funding

| Funder | Grant reference number | Author |
|---|---|---|
| National Institute of Neurological Disorders and Stroke | 1R21NS093540-01 | Mary E Hatten |
| Rockefeller University | Pilot award | Hourinaz Behesti Mary E Hatten |
| Starr Foundation | Tri-Institutional Stem Cell Initiative Grant | Mary E Hatten |
| US Army Medical Research Acquisition Activity | W81XWH1510189 | Mary E Hatten |
| Rockefeller University | | Mary E Hatten |
| Renate, Hans, and Maria Hofmann Trust | | Mary E Hatten |

| Funder | Grant reference number | Author |
| --- | --- | --- |

The funders had no role in study design, data collection and interpretation, or the decision to submit the work for publication.

## Author contributions

Hourinaz Behesti, Conceptualization, Data curation, Formal analysis, Methodology, Validation, Visualization, Writing - original draft; Arif Kocabas, Investigation, Validation; David E Buchholz, Consultation, Resources; Thomas S Carroll, Data curation, Formal analysis, Visualization; Mary E Hatten, Conceptualization, Funding acquisition, Writing – review and editing

## Author ORCIDs

Hourinaz Behesti (iD) http://orcid.org/0000-0001-9383-9929
David E Buchholz (iD) http://orcid.org/0000-0003-4021-7696
Mary E Hatten (iD) http://orcid.org/0000-0001-9059-660X

## Ethics

Human subjects: Fixed de-identified human tissue were acquired from the Human Developmental Biology Resource (http://www.hdbr.org/) following institutional policies.
This study was performed in strict accordance with the recommendations in the Guide for the Care and Use of Laboratory Animals of the National Institutes of Health. All of the animals were handled according to approved institutional animal care and use committee (IACUC) protocol (#14746-H) of the Rockefeller University. All surgery was performed under hypothermia, and every effort was made to minimize suffering.

## Decision letter and Author response

Decision letter https://doi.org/10.7554/eLife.67074.sa1
Author response https://doi.org/10.7554/eLife.67074.sa2

# Additional files

## Supplementary files

• Supplementary file 1. GO terms for TRAP-seq data.
• Transparent reporting form

## Data availability

Sequencing data have been deposited in GEO under accession code: GSE163710.

The following dataset was generated:

| Author(s) | Year | Dataset title | Dataset URL | Database and Identifier |
| --- | --- | --- | --- | --- |
| Behesti H, Hatten ME, Kocabas A, Carroll TS | 2020 | TRAP seq of the human pluripotent stem cell derived ATOH1 lineage | https://www.ncbi.nlm.nih.gov/geo/query/acc.cgi?acc=GSE163710 | NCBI Gene Expression Omnibus, GSE163710 |

The following previously published datasets were used:

| Author(s) | Year | Dataset title | Dataset URL | Database and Identifier |
|---|---|---|---|---|
| Li M, Santpere G, Kawasawa YI, Evgrafov OV, Gulden FO, Pochareddy S, Sunkin SM, Li Z, Shin Y, Zhu Y, Sousa AMM, Werling DM, Kitchen RR, Kang HJ, Pletikos M, Choi J, Muchnik S, Xu X, Wang D, Lorente-Galdos B, Liu S, Giusti-Rodríguez P, Won H, Leeuw CAde, Pardiñas AF, BrainSpan Consortium, PsychENCODE Consortium, PsychENCODE Developmental Subgroup, Hu M, Jin F, Li Y, Owen MJ, O'Donovan MC, Walters JTR, Posthuma D, Reimers MA, Levitt P, Weinberger DR, Hyde TM, Kleinman JE, Geschwind DH, Hawrylycz MJ, State MW, Sanders SJ, Sullivan PF, Gerstein MB, Lein ES, Knowles JA, Sestan N | 2018 | Integrative functional genomic analysis of human brain development and neuropsychiatric risks | http://development.psychencode.org | Human mRNA seq processed data: Gene expression in counts, psychencode |
| Wizeman JW, Guo Q, Wilion EM | 2019 | Specification of diverse cell types during early neurogenesis of the mouse cerebellum | https://elifesciences.org/articles/42388 | elifesciences, 10.7554/eLife.42388.018 |

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
