## [Editor Report]

Your paper beautifully addresses the divergent mechanisms that create species-specific features of brain development, focusing on transcriptional programs and their timing, that generate human cerebellar cells compared to those in mouse. You describe a rapid protocol for the derivation of the human ATOH1 lineage that generates excitatory cerebellar neurons from human embryonic stem cells (hESCs), and study them in vitro and in vivo. You observed transcription factors classically associated with mouse differentiated neurons expressed in the human outer external granule layer where granule cell precursors reside. These results argue that the prolonged development of the cerebellum in the human is linked to its increased size in evolution.

---

## [Decision Letter]

**Decision letter after peer review:**

Thank you for submitting your article "Altered temporal sequence of transcriptional regulators in the generation of human cerebellar granule cells" for consideration by *eLife*. Your article has been reviewed by 3 peer reviewers, and the evaluation has been overseen by a Reviewing Editor and Catherine Dulac as the Senior Editor. The following individuals involved in review of your submission have agreed to reveal their identity: Noriyuki Koibuchi (Reviewer #2); Jason Tchieu (Reviewer #3).

Essential revisions:

The Reviewers thought your study was a valuable addition to stem cell models, and to human cerebellar development. The presentation of your results are very detailed, but the reviewers had equally detailed questions and comments, most of which could be addressed textually in the rebuttal and/or main Text and would bolster the impact of your study.

There are five areas in which the reviewers recommended amendment:

(1) Characterization of the more mature cell types:

a. Although you can consistently generate GCP-like cells with your protocol, can you provide more data on the identity of the cells that do not express the canonical markers, in particular cells negative for En2 and/or for Atoh1? Similarly, have you checked for markers of differentiated GC other than TAG1? These data would greatly strengthen the notion that your method is improved compared to previously described approaches.

b. The method used to differentiate hESCs to ATOH1+ cells is highly efficient; however, a low proportion (~13%) of the cells become TAG1+ by d28. Are the TAG1- cells still ATOH1+ in the flow through? Are these cells stuck at the patterning stage, and do you think that these cells will become TAG1+ later? Will the flow through cells become brush cells or other neurons of the cerebellum? Or did these cells lose the identity of the cerebellar territory? This should be discussed.

c. Reviewer 2 points to Fig. 1E: although you state that differentiation starts by DIV23 due to the "decrease" in ATOH1-EGFP positive cells and "increase" in ATOH1-EGFP negative cells, such changes are not clear. The changes do not seem to be statistically significant. This concern may be clearer on DIV28, when a great SD in the ratio of ATOH1-EGFP negative cells with slightly higher ratio of ATOH1-EGFP positive cells than that in DIV23, indicating that differentiation may not be in progress. Furthermore, in Figure 1 supplement E, ATOH1 levels increased in DIV23. Such increase may not be consistent with findings and comments on Fig. 1E.

d. Similarly, Reviewer 3 comments that since the work was largely performed on ATOH1-enhancer driving nuclear GFP, the results will likely need additional data/explanation to support the out-of-order development/differentiation. It would be important to determine if the %ATOH1-GFP+ cells indeed express ATOH1 as well as the markers NEUROD1 and RBFOX3 to demonstrate overlap on a cellular level. Is there any point during the development of the mouse cerebellum where there is potential overlap of the post-mitotic markers with ATOH1? Moreover, would your previous transplantation experiments (Figure 2G) be able to shed light on human and mouse specific development in terms of timing?

(2) Whether the behavior/localization of the transplanted cells accurately indicates cell fate:

The transplantation experiments are a very nice addition, but from the data shown it is unclear whether genuine glia-directed migration has occurred here. Apart from performing time-lapse imaging (not required here!), could you provide more detail or data immediately after transplantation – with examples of the site of injection, and compare these images to where the cells end up 48 hrs later? It would be welcome to determine, or provide any data you may have, on whether the transplanted cells proliferate or differentiate further (using same markers as those used in your in vitro or in your or other in vivo studies, and if they survive in the long run in the mouse cerebellum (rough indication of the numbers of cells injected with those that integrate and survive)).

(3) Discussion of evidence from your data or other published data that to fortify the difference in species-specific timing:

Your evidence implicating heterochronic (precocious) expression of neuronal markers in the human GCP is novel and exciting but it should be examined further to back up the your interpretation. Firstly, you should check on other stages of human development to make sure that the choice of comparing P0 mouse with PCW17 human is indeed valid and meaningful, and that the decreased proliferative patterns described in the human reflect a genuine species difference and not contrasting stages of development. You have taken advantage of the Psychencode data but have you consulted RNAseq data on mouse cerebellar development (Carter et al 2018) to further compare your own datasets, as well as assess whether NeuroD1 and NeuN are indeed absent from mouse GCP at relevant stages. These efforts would reveal additional information on timeline comparison between human and mouse cerebellar neurogenesis.

(4) More consistent statistical analysis:

a. There is little comment on statistical difference in each dataset in the Results section, although you state "increase" or "decrease" for each data group.

b. In relation to the comment above, in Fig. 1E, to verify a change in ATOH1-EGFP positive negative cells by DIV23, statistics should be shown. Furthermore, in Figure 1 supplement E, ATOH1 level increased at DIV23. If differentiation was in progress before DIV 23, ATOH1 levels would decrease. Please describe more clearly why you think that differentiation starts at DIV23.

(5) Details of the in vitro setting:

a. The authors devise a protocol to generate excitatory cerebellar neurons from hESCs. It appears there are 3 phases in this differentiation: (1) regional patterning, (2) lineage specification and (3) maturation. It is a bit confusing to follow the rationale for each step in the text as it appears to jump back and forth from patterning and specification. For example, the focus on the detailed patterning of the cerebral territory jumps to efficiency of ATOH1 unexpectedly. Clarification of this aspect would improve the overall understanding during the optimization of this protocol.

b. During patterning: the authors want EN2+, GBX2+, PAX6- and OTX2- progenitors and from Supplemental Figure 1B, it appears that condition 7 and 8 are ideal. It would be more convincing to rerun the OTX2 portion of this gel, or provide further information, to confirm that it is not present in these two conditions and not just due to a darker contrast.

c. You make statements that both FGF8b and FGF2 are equivalent and that FGF2 promotes the survival, high levels of BMP7 induces cell death and BDNF improves survival. Do you have data to support these statements?

d. Please comment on how you defined the optimal conditions for administering BDNF. Also, add any data on whether BDNF treatment alters the expression of ATOH1 and other markers such as EN2, MEIS2 and GBX2.

e. The transwell differentiation is interesting and sounds promising as it appears to reduce the variability seen in ordinary culture dishes. It remains unclear whether the transwell helps the patterning or just for reaching the ATOH1 stage. You demonstrate that a lower density of plating (900 cells/ml) leads to reduced variability and great efficiency to obtain ATOH1+ cells. How many ATOH1+ cells do you obtain per hESC? How many become TAG1+? And how many ultimately become GCs? This will help assess the utility of the differentiation strategy.

*Reviewer #1:*

Behesti et al. describe a new protocol of generation of cerebellar cells, focusing on granule cell progenitors (GCP), derived from human PSC. They show that the addition of GSK3 inhibitor CHIR along with FGF2 is more efficient than the previously reported FGF-insulin treatment to induce cerebellar markers and repress midbrain markers. Moreover, the differentiation protocol highlights the importance of introducing BMP7 to the in vitro culture, as well as placing the cells on transwell membranes in order to decrease the variance of ATOH1 induction. The resulting GCP can be transplanted in neonatal cerebellum, where they show evidence suggestive of glia-directed migration. Then by using TRAPseq, they compare the transcriptome of PSC-derived GCP from their system with publicly available RNAseq data of human cerebellum, which shows highest similarity between PSC-derived cells and early- to mid-gestational cerebellum. Finally, the authors report expression of NeuroD1 and RBFOX3 in the human GCP, both in the in vitro system and in vivo at mid-fetal stages, while these factors of neuronal differentiation are absent in the mouse GCP. Conversely they show reduced expression of Ki67 marker of proliferation in the human cells, which leads the authors to suggest that the apparent heterochrony of expression of NeuroD1/NeuN could underly the longer maintenance of GCP in human fetal cerebellum, in line with the prolonged cerebellar neurogenesis in the human until 1-2 years after birth. This is a potentially interesting study that carefully describes an enhanced method to generate human GCP, an important cell type in human brain development, evolution and disease. The expression data on species-specific timing of expression of NeuroD1 and NeuN are novel and potentially exciting, although their exact significance remains to be explored further.

*Reviewer #2:*

Behesti et al tried to develop in vitro system to generate human cerebellar granule cells from human pluripotent stem cells (hPSC). First, they defined optimal conditions for deriving human cerebellar territory from hPSC by carefully adding different compounds with various timing. Then they developed a clonal hPSC line expressing EGFP under enhancer of ATOH1, which is an essential transcription factor for cerebellar neurogenesis. Using this cell line with their defined culture conditions, they successfully purified ATOH1+ cells with a high yield by FAC sorting. Then they further purified cells harboring postmitotic granule cell character using Magnetic Activated Cell Sorting using an antibody against TAG1, which is expressed in postmitotic granule cells located in the premigratory zone. These cells can grow in co-culture with mouse glial cells and undergo cell migration in the mouse cerebellum. Then using translating ribosome affinity purification (TRAP) methodology, they performed a transcriptional profiling and found that DIV28 ATOH1 lineage most closely matched the profile of the PCW13-17 human cerebellum. This in vitro system can be a useful tool to study the development of human granule cells.

A strength is that their culture condition was carefully examined and they successfully found an optimal condition to derive a human cerebellar territory. They also generated transgenic cell line expressing EGFP under ATOH1 enhancer. Using this cell line under optimal culture condition with various sophisticated techniques, they produced ATOH1 positive cells within shorter DIV with higher yield compared with a previous study. Using the TRAP method, they performed transcriptional profiling in a cell lineage-specific manner. Such profiling allows them to compare in detail the resemblance between human expression profiling in the database and their ATOH1 positive cells. This comparison is useful to confirm that they successfully developed cerebellar granule cells in vitro. A weakness is that although the authors stated in the Methods section that they performed statistical analysis, comments on statistical differences in each data set are limited in the Results section, although they stated an "increase" or "decrease" for each data set.

*Reviewer #3:*

Human cerebellar development is a tightly regulated process which has important implications in nervous system disorders and cancer however methods to study these cells is limited. The manuscript by Behesti et al. developed a novel monolayer method to generate rhombic lip derived cerebellar granule cells (GC) from human embryonic stem cells (hESCs). Interestingly, dosage of BMP7 appears critical in reducing the variability of the differentiation. The authors generated ATOH1-GFP hESC lines and performed TRAP-sequencing to identify the similarities of the in vitro derived cells to in vivo expression. Transcriptomic analysis of the GC progenitors suggests that after 28 days of differentiation, these cells correlate best with a cerebellum profile in the second trimester. Surprisingly, expression of neuron-specific factors in the GC progenitors in human derived cells suggest a difference between human and mouse cerebellar development and was confirmed in human tissue.

The data presented in this manuscript are interesting even though there are a number of alternative cerebellar differentiation protocols. The highlight of this method is the strong focus on recapitulating development in a stepwise manner to generate the granule neurons with careful titration of WNT, FGF and BMP7 to fine tune expression of ATOH1. The data supporting the identity of the in vitro derived cells is convincing. However, there are some aspects of the manuscript where additional data would be necessary to prevent misinterpretation of the results.

[Editors' note: further revisions were suggested prior to acceptance, as described below.]

Thank you for resubmitting your work entitled "Altered temporal sequence of transcriptional regulators in the generation of human cerebellar granule cells" for further consideration by *eLife*. Your revised article has been reviewed by 3 peer reviewers and the evaluation has been overseen by Catherine Dulac as the Senior Editor, and a Reviewing Editor.

Your paper addresses the divergent mechanisms that create species-specific features of brain development, focusing on transcriptional programs, and their timing, that generate human cerebellar cells compared to those in mouse. You describe a rapid protocol for the derivation of the human ATOH1 lineage that generates excitatory cerebellar neurons from human embryonic stem cells (hESCs). You then developed a clonal hPSC line expressing EGFP under enhancer of ATOH1, enabling FAC sorting of these cells and further cell purification with an antibody against TAG1, expressed in postmitotic granule cells located in the premigratory zone, You successfully co-cultured these cells in a trans-well configuration with mouse glial cells, and observed migration of the human-derived cells after transplantation into the mouse cerebellum.

Transcriptional profiling then indicated that the mouse postnatal ATOH1 lineage most closely matched the profile of the embryonic human cerebellum, with transcription factors classically associated with differentiated neurons in mouse situated in the human outer external granule layer housing granule cell precursors. Your results suggest mechanisms underlying the prolonged development of the cerebellum in the human that may be linked to its increased size in evolution.

All three reviewers agreed that the manuscript has been greatly improved but there are some remaining issues that need to be addressed, as outlined below:

Reviewer 1 asked for two further revisions:

1. As NeuroD1 is expressed in other brain regions than cerebellum (including cortex and other forebrain regions cf for instance scrnaseq data on ucsc from Kriegstein lab) it would be very valuable to know the proportion of NeuroD1/Pax6 double positive cells (instead of each marker separately) as this combination is much more specific. Perhaps elaborate on your rebuttal comment #1.

2. In absence of more data on human time course of expression of NeuroD1 and other markers, the authors should tone down their conclusion on heterochrony – rather describe it as divergent pattern of expression that could be consistent with heterochrony.

3. From the Reviewing Editor: with regard to differences in overall timing, it is most interesting that Sox2 is expressed in the human EGL unlike in mouse. You say in your rebuttal "we show that a surprisingly large number of Sox2^+^ cells are present in the human EGL (Figure 5. Figure supplement 1D). Moreover, in our cultures, 72% +/-16 of the Sox2^+^ cells are ATOH1-EGFP+ (GCP identity), while 27% of the Sox2^+^ cells are EGFP-…." Although you added a paragraph to page 6 to describe these data, it would be welcome to reiterate this in the Discussion to forify the idea that the cells in the human EGL are "held" in an extended immature (stem cell) state as well as heterochronically expressing transcription factors that in mouse are expressed at later stages.

---

## [Author Response]

Essential revisions:The Reviewers thought your study was a valuable addition to stem cell models, and to human cerebellar development. The presentation of your results are very detailed, but the reviewers had equally detailed questions and comments, most of which could be addressed textually in the rebuttal and/or main Text and would bolster the impact of your study.There are five areas in which the reviewers recommended amendment:(1) Characterization of the more mature cell types:a. Although you can consistently generate GCP-like cells with your protocol, can you provide more data on the identity of the cells that do not express the canonical markers, in particular cells negative for En2 and/or for Atoh1? Similarly, have you checked for markers of differentiated GC other than TAG1?

In the initial submission of the manuscript, we showed MAP2, NeuroD1, and Synaptophysin in addition to TAG1 expression in Tag1+ cells isolated at DIV28 and grown an additional 20 days in co-culture with mouse cerebellar neurons and glia. We also described the characteristic small round morphology of differentiated granule cells, which is not apparent when they are still proliferating or migrating. See Fig. 2E-F. In the brain, NeuroD1 expression is largely localized to cerebellar granule cells as well as in granule cells in the hippocampus and hence, it is a fairly selective granule cell marker. We have now performed additional analyses including the quantification of NeuroD1+ and Pax6+ cells at DIV28+20 and quantification of cells with a small round nucleus. The text on p.7 second paragraph has been updated with this information. We have also added the expression of VGLUT1, a glutamatergic synaptic marker, expressed in cerebellar granule cell parallel fibers (see new Fig. 2G) as an additional marker of differentiated GCs.

To address the question of the identity of cells negative for ATOH1 (~20% of the cells in cultures at DIV28), we carried out further double immunolabeling experiments at two timepoints:

1. At DIV28 using antibodies against PAX6, NeuN, Calretinin, and SOX2 (in combination with ATOH1-EGFP). We have evidence that the great majority of the cells are rhombic lip (RL) and RL derivatives. Evidence to support this conclusion includes a large number of cells that expressed PAX6 (which is expressed both in the EGL and the RL), see (Haldipur et al., 2019), a proportion of which are double positive for ATOH1-EGFP^+^;NeuN^+^ (see new Fig. 1 – Supplementary Fig. 3B). ATOH1 and NeuN are expressed in human GCPs (our finding) but are largely absent from RL cells (Haldipur et al., 2019) and hence regarding your question about the identity of ATOH1^-^ cells, our data suggest that they are mostly RL cells (PAX6+; NeuN-) with smaller contributions of differentiated GCs (PAX6+; NEUN+; ATOH1-EGFP-), cerebellar nuclei and/or Unipolar brush cells (Calretinin+, see new Fig. 1 figure supplement 3). We also examined SOX2 expression. SOX2 is typically expressed in the ventricular zone neuroepithelium, which is ATOH1^-^ and gives rise to the GABAergic lineage and glial progenitors. A small number of Sox2^+^ cells have been reported in the mouse EGL, although they constitute a rare population (Selvadurai et al., 2020; Sutter et al., 2010). Here, by immunohistochemistry, we show that a surprisingly large number of Sox2^+^ cells are present in the human EGL (Figure 5. Figure supplement 1D). Moreover, in our cultures, 72% +/-16 of the Sox2^+^ cells are ATOH1-EGFP+ (GCP identity), while 27% of the Sox2^+^ cells are EGFP-, suggesting that these cells (27%) may represent glial progenitors/ventricular zone progenitors.

We have added a paragraph to page 6 to describe these data.

2. At DIV28+20, to estimate the proportion of cells that will go on to a unipolar brush cell or cerebellar nuclei identify (Calretinin+/NeuN+/-), we quantified the presence of these markers in the TAG1^-^ fraction after 20 days in culture post sorting (DIV28+20) and observed again that Calretinin+ cells constitute only a small fraction of the population (~5%). We have added a paragraph on Page 8 to describe these data.

Together, our analysis suggests that at DIV28 our cultures contain mostly RL cells and GCPs and a small number of cerebellar nuclei/unipolar brush cells, in addition to newly postmitotic GCs marked by Tag1 and PAX6+;NeuN+ (EGFP-/EN2+/-). We cannot exclude that some of the cells in culture may have brain stem identity. Hence purification steps such as Tag1 sorting at different time-points provides a strategy to obtain more pure populations of a particular cell type of interest.

These data would greatly strengthen the notion that your method is improved compared to previously described approaches.b. The method used to differentiate hESCs to ATOH1+ cells is highly efficient; however, a low proportion (~13%) of the cells become TAG1+ by d28. Are the TAG1- cells still ATOH1+ in the flow through?

Yes, we still see a large number of ATOH1+ cells in the TAG1- fraction after purification. This is because TAG1 is a transient marker that is switched on and then off before the migration of newly postmitotic GCs. Hence, at DIV28, the 13% reflects the population that have transiently switched on TAG1 at that moment in time. In the TAG1- fraction, other cells will eventually switch on and switch off TAG1 and go on to give rise to granule cells, mimicking the extended period of granule cell genesis from the EGL, observed both in the mouse and human cerebella. This is further supported by the fact that we have successfully isolated TAG1+ cells that go on to show granule cell marker/morphology at DIV35 (data not shown) with a similar yield to DIV28 (~13%). We have added a few sentences to clarify these points on P7 and P8.

Are these cells stuck at the patterning stage, and do you think that these cells will become TAG1+ later?

Some do become TAG1+ as described above and a great majority go on to express granule cell markers. Please see new data presented in Fig 2. Supplementary Fig 1C.

Will the flow through cells become brush cells or other neurons of the cerebellum? Or did these cells lose the identity of the cerebellar territory? This should be discussed.

As discussed above, a subset of the flow-through go on to become granule cells (Fig 2. Supplementary Fig 1A, C). However, at DIV28, as stated earlier, we have a mixed population of cells that we previously showed contain at least 2 of the three excitatory neuronal cell types, the cerebellar nuclei (Calretinin+) and GCs. To address whether there are also differentiated GCs and unipolar brush cells/cerebellar nuclei (Calretinin+), we looked at PAX6+/NeuN+/ATOH1EGFP- (differentiated GCs) and Calretinin. Only ~5% of the cells were Calretinin+, but we detected a large number of cells expressing granule cell markers. Please see Figure 2 – figure supplement 1C for quantifications of the other markers.

c. Reviewer 2 points to Fig. 1E: although you state that differentiation starts by DIV23 due to the "decrease" in ATOH1-EGFP positive cells and "increase" in ATOH1-EGFP negative cells, such changes are not clear. The changes do not seem to be statistically significant. This concern may be clearer on DIV28, when a great SD in the ratio of ATOH1-EGFP negative cells with slightly higher ratio of ATOH1-EGFP positive cells than that in DIV23, indicating that differentiation may not be in progress. Furthermore, in Figure 1 supplement E, ATOH1 levels increased in DIV23. Such increase may not be consistent with findings and comments on Fig. 1E.

The differences are small as pointed out by the reviewers, but the graph does show a trend that we have consistently observed; ATOH1-EGFP+ cell numbers go down around DIV23.

Moreover, our conclusion that differentiation picks up from DIV23 onwards is based on multiple observations. We would like to draw the attention of the reviewers to figure 2A where we had shown the appearance of TAG1, an early differentiation marker already at DIV 18 in cultures with a marked increase by DIV23. Also, we would like to point out that the analysis shown in Fig. 1- figure supplement 1 F (formerly 1E) is based on gene expression in ATOH1-EGFP-sorted cells, not total cells, i.e.: not the differentiated proportion. Hence, the fact that ATOH1 expression goes up in the ATOH1-EGFP+ cells is not incompatible with there being differentiation in the cultures as a whole. As requested, we analyzed the data in Fig. 1E by ANOVA and have provided the result in the figure legend.

d. Similarly, Reviewer 3 comments that since the work was largely performed on ATOH1-enhancer driving nuclear GFP, the results will likely need additional data/explanation to support the out-of-order development/differentiation. It would be important to determine if the %ATOH1-GFP+ cells indeed express ATOH1 as well as the markers NEUROD1 and RBFOX3 to demonstrate overlap on a cellular level.

The out of order development, marked by an altered sequence of expression of transcriptional regulators in the human cells is supported by several pieces of data. First, we provide further evidence of ATOH1-EGFP^+^;Neun^+^;NeuroD1^+^ cells in our cultures as requested by the reviewer (see Fig.1 – figure supplement 3B). This is consistent with our previously reported observation of the expression of these factors in the outer EGL in the developing human cerebellum on sections (Fig. 5).

Unfortunately, none of the available ATOH1 antibodies worked on human sections or human cells and hence we are unable to provide direct evidence of ATOH1 expression at protein level. However, as previously shown in Fig. 2 – figure supplement 2A, B, we see enrichment of ATOH1 transcripts in the ATOH1-EGFP+ populations of our transgenic lines (both ATOH1-EGFP and ATOH1-EGFP-L10a). This together with the co-expression of a number of granule cell lineage markers in these cells in vitro (PAX6, NEUN, NEUROD1), and the fact that we detect early onset of expression of NEUN and NEUROD1 in the human outer EGL, which is spatially identifiable on human cerebellar sections, we believe provide strong evidence of an altered sequence of these transcriptional regulators in the developing human cerebellum, which we have succeeded in modeling in vitro.

Moreover, we provide new data on an additional marker, SOX2 in the human EGL, that is different to its expression pattern in the mouse. Previous studies show that Sox2^+^ cells are a rare find in the mouse EGL (Selvadurai et al., 2020; Sutter et al., 2010), but we detected many Sox2^+^ cells throughout the human EGL. This finding is supported by two recent reports on the expression of SOX2 in the developing human EGL (Pibiri et al., 2016; Selvadurai et al., 2020). We show that SOX2 is largely coexpressed with ATOH1-EGFP in our cultures. The ATOH1-EGFP+;Sox2^+^ cells constitute ~72% of the Sox2^+^ population (Fig.1 – figure supplement 3C, D).

Is there any point during the development of the mouse cerebellum where there is potential overlap of the post-mitotic markers with ATOH1? Moreover, would your previous transplantation experiments (Figure 2G) be able to shed light on human and mouse specific development in terms of timing?

As our transplantation experiments lasted only 48 hrs, this was too short of a window to be able to assess the differentiation state of the cells, but please see further analyses and explanations above to the previous questions. In future experiments, it will indeed be interesting to follow the fate of the cells after longer periods of development in the transplanted mouse brain. To address if there are any points during the development of the mouse cerebellum when ATOH1 overlaps with post-mitotic markers, we added one more embryonic timepoint at E17.5 and one more postnatal timepoint (P6) to our analyses to cover the window of cerebellar development in the mouse from E15.5 to P6. We never observed NeuN expression in the mouse outer EGL at any time point examined and NeuroD1 was also largely absent from the outer EGL at most timepoints in the mouse. At P6 however, the number of NeuroD1+ cells at the pial surface were higher compared to other timepoints in the mouse. P6 is a relatively late stage of cerebellar development in the mouse, when GCP proliferation peaks (Ki67 throughout the EGL) and the EGL is at its thickest. It has previously been reported that this postnatal stage in the mouse is most comparable to weeks 28-34 in the human developing cerebellum, based on when peak proliferation and thickness are observed in the developing human EGL (Abraham et al., 2001). This is indeed in-line with our assessment that the 17PCW cerebellum most closely resembles an earlier stage in the mouse (P0) based on EGL thickness and morphology.

We have modified the text on p10 to reflect this new analysis.

(2) Whether the behavior/localization of the transplanted cells accurately indicates cell fate:The transplantation experiments are a very nice addition, but from the data shown it is unclear whether genuine glia-directed migration has occurred here. Apart from performing time-lapse imaging (not required here!), could you provide more detail or data immediately after transplantation – with examples of the site of injection, and compare these images to where the cells end up 48 hrs later? It would be welcome to determine, or provide any data you may have, on whether the transplanted cells proliferate or differentiate further (using same markers as those used in your in vitro or in your or other in vivo studies, and if they survive in the long run in the mouse cerebellum (rough indication of the numbers of cells injected with those that integrate and survive)).

To address the reviewers’ concern about whether our in vivo data show genuine glia-directed migration, we performed additional experiments. We assumed that this comment stemmed from alternative interpretations, such as: did the human cells in the molecular layer and the IGL end up there because they were possibly transplanted there in the first place (instead of migrating there) or whether they entered the molecular layer by means other than glial guided migration (such as migration along blood vessels or other cells present in the environment).

To address this, we performed in vitro cultures of human TAG1-sorted neurons (i.e. early postmitotic neurons) with mouse glia, re-creating the classical co-culture experiments where glial-guided neuronal migration of cerebellar granule cells were first studied in vitro (Hatten, 1985). This in vitro system gets rid of other cell types (such as endothelial cells) and allows direct visualization of neuronal attachment on glia and provides a reductionist cellular environment, where other cell types/structures such as blood vessels are no longer present near imaged neurons. We show evidence that the human neurons (HuNU+, MAP2+) attach to glia and show the characteristic elongated nuclear shape of migrating neurons (Fig. 3B), supporting the conclusions from our in vivo transplantations that the hESC-derived cerebellar neurons can indeed migrate along glia.

We also provide images that show that the great majority of the transplanted cells were still situated on the pial surface of the cerebellum at 48 hrs after transplantation (Fig. 3- figure supplement 1). This suggests that most neurons have yet to integrate but that the ones observed in the molecular layer and in the IGL have indeed migrated there and were not transplanted there. In future work, it would indeed be interesting to analyze later time-points to determine if a larger fraction of the cells enter the cerebellar cortex and the integration/differentiation percentage of the transplanted cells.

(3) Discussion of evidence from your data or other published data that to fortify the difference in species-specific timing:Your evidence implicating heterochronic (precocious) expression of neuronal markers in the human GCP is novel and exciting but it should be examined further to back up the your interpretation. Firstly, you should check on other stages of human development to make sure that the choice of comparing P0 mouse with PCW17 human is indeed valid and meaningful, and that the decreased proliferative patterns described in the human reflect a genuine species difference and not contrasting stages of development.

To address this comment and provide further evidence as requested, we undertook two additional approaches. 1) As obtaining more human fetal samples, suggested by the reviewers, was an obstacle, we instead added two more time-points to our previous data in the mouse and expanded our analysis of the mouse, covering a large window of EGL development (E15.5, E17.5, P0, P6). Our data show that, in contrast to our observation in the human, NeuN is not detected at any of these time points in the outer-EGL or the pial surface in the developing mouse cerebellum, corroborating our previous findings. While NeuroD1+ cells were occasionally detected at the pial surface in the mouse cerebellum as reported in our first submission, the percentage of NeuroD1+ cells at the pial surface were lower at all time-points in the mouse compared to human except at P6, where we detect similar numbers to the human overall. However, morphologically (foliation, lamination and EGL thickness), the human 17PCW is far more immature appearing than the mouse P6 (see also (Abraham et al., 2001; Haldipur et al., 2019) and as stated previously, is most alike a P0 mouse cerebellum. Hence our data corroborate our conclusion that the sequence of expression of transcriptional regulators in the developing human cerebellum is altered compared to the mouse and that the transcriptional and morphological maturity in the two species do not align, with human cells displaying a far more mature transcriptional signature than the morphological maturity of the cerebellar structure.

You have taken advantage of the Psychencode data but have you consulted RNAseq data on mouse cerebellar development (Carter et al 2018) to further compare your own datasets, as well as assess whether NeuroD1 and NeuN are indeed absent from mouse GCP at relevant stages. These efforts would reveal additional information on timeline comparison between human and mouse cerebellar neurogenesis.

We were unable to perform gene set analysis comparison of our data with Carter et al. as the publicly available files are raw unprocessed single cell RNA-seq data and we were unable to access the processed files, which would have facilitated such a comparison without the need to re-analyze all their data. We did however carry out gene set enrichment analysis of our data compared to another processed single cell RNA seq study of the developing mouse cerebellum (Wizeman et al., 2019), which had incorporated parts of the Carter et al data set. This analysis showed that our data most closely aligned with the glutamatergic lineages, followed by the GABAergic lineage and importantly did not align with data from glia/other non-neuronal cell types. We have provided a new figure to display the comparison of our data to the cell-types identified in Wizeman et al. 2019 in new Fig. 4 -figure supplement 2. It should be noted that a number of genes that are also associated with the glutamatergic lineages, such as Sox2 and NeuroD1, also appeared in their GABAergic categories (See “Wizeman data” in Figure 4 – source data 3 for genes/category). We did not re-classify/re-analyze their data, but simply performed a comparative analysis to their cell categories as was. Moreover, manual interrogation of both the Wizeman et al and Carter et al data showed that NeuroD1 is co-expressed in a minority subset of ATOH1 positive cells in mouse cerebellum, which support our immunohistochemistry results. Rbfox3 (NeuN) appears largely non-overlapping with Atoh1, again corroborating our data and previous reports.

(4) More consistent statistical analysis:a. There is little comment on statistical difference in each dataset in the Results section, although you state "increase" or "decrease" for each data group.

As requested, we have now provided more statistics, although previously we had accurately described trends in our data as observed “increase” and “decrease” and never stated that anything was “significantly” increased or decreased. Most of our data are observational and not comparative in nature.

b. In relation to the comment above, in Fig. 1E, to verify a change in ATOH1-EGFP positive negative cells by DIV23, statistics should be shown. Furthermore, in Figure 1 supplement E, ATOH1 level increased at DIV23. If differentiation was in progress before DIV 23, ATOH1 levels would decrease. Please describe more clearly why you think that differentiation starts at DIV23.

The gene expression analysis (Figure 1—figure supplement 1 F, formerly E) was carried out in FAC-sorted ATOH1-EGFP+ cells, not in total cells, as stated earlier. Hence the increase in expression merely reflects the increased levels of ATOH1 transcript in ATOH1-EGFP+ cells, which is not incompatible with differentiation of other cells in the cultures. We have modified the sentence on Page 5 line 22 to clarify this point.

We have compared the data in Fig. 1E by ANOVA (described in methods and in the fig legend).

(5) Details of the in vitro setting:a. The authors devise a protocol to generate excitatory cerebellar neurons from hESCs. It appears there are 3 phases in this differentiation: (1) regional patterning, (2) lineage specification and (3) maturation. It is a bit confusing to follow the rationale for each step in the text as it appears to jump back and forth from patterning and specification. For example, the focus on the detailed patterning of the cerebral territory jumps to efficiency of ATOH1 unexpectedly. Clarification of this aspect would improve the overall understanding during the optimization of this protocol.

We have added a sentence on page 5 line 3 to better describe the rationale for jumping from cerebellar territory to GCP specification and modified wording in the manuscript with this comment in mind in multiple other sentences (all highlighted in red).

b. During patterning: the authors want EN2+, GBX2+, PAX6- and OTX2- progenitors and from Supplemental Figure 1B, it appears that condition 7 and 8 are ideal. It would be more convincing to rerun the OTX2 portion of this gel, or provide further information, to confirm that it is not present in these two conditions and not just due to a darker contrast.

We have re-run this PCR and gel as requested and modified the figure with an image of the new gel. The results for conditions 7 and 8 were the same as the previously shown gel. Moreover, we would like to point out that we show 2 other gels where OTX2 is clearly not expressed in condition 8, Fig. 1 -supplementary fig 1A and D.

c. You make statements that both FGF8b and FGF2 are equivalent and that FGF2 promotes the survival, high levels of BMP7 induces cell death and BDNF improves survival. Do you have data to support these statements?

We have added images of ATOH1-EGFP and TAG1 expression in cultures treated with FGF8 versus FGF2 as an indication of cell survival (Figure 1. Figure supplement 1C). In data not shown, we consistently observed that the color of the medium in FGF2 treated wells were more yellow than the medium in FGf8 treated wells, which indicates that there are more cells consuming the medium in FGF2 wells and hence better survival. In the text we had indicated that BDNF was added to improve GC survival and this was based on published work in the mouse that had shown that BDNF significantly improves granule cell survival (cited in the text, P5 line 11). BDNF is routinely included in most stem cell differentiation protocols to improve cell survival and hence we added BDNF based on this knowledge at concentrations routinely used.

d. Please comment on how you defined the optimal conditions for administering BDNF. Also, add any data on whether BDNF treatment alters the expression of ATOH1 and other markers such as EN2, MEIS2 and GBX2.

The expression of the above markers is already established prior to the addition of BDNF at DIV11 (See Fig. 1 -supplementary fig 1). Hence BDNF does not play a role in the early patterning of the cultures. The BDNF concentration was based on previously reported concentrations used in mouse cultures to significantly improve granule cell survival (Lindholm et al., 1993).

e. The transwell differentiation is interesting and sounds promising as it appears to reduce the variability seen in ordinary culture dishes. It remains unclear whether the transwell helps the patterning or just for reaching the ATOH1 stage. You demonstrate that a lower density of plating (900 cells/ml) leads to reduced variability and great efficiency to obtain ATOH1+ cells.

The transwells helped improve overall cell survival and stabilize ATOH1 expression mainly. All other markers were consistently detected in our experiments regardless of culture surface. Hence the transwells do not alter the initial overall patterning of the cultures.

How many ATOH1+ cells do you obtain per hESC? How many become TAG1+? And how many ultimately become GCs? This will help assess the utility of the differentiation strategy.

As TAG1 is a transient marker and granule cell genesis is a continuous process in these cultures we can only give estimates at snapshots in time and not an absolute number of how many GCs are produced per hESC. We have continued these cultures up until day 70 in vitro without TAG1 sorting and they continue to show ATOH1-EGFP positive cells at that stage, reflecting the continued source of GCPs. As mentioned previously, at DIV28, 13% are TAG1+ and by DIV 28+20 in the TAG1+ fraction, 81% are PAX6+ and 75% are NEUROD1+ with 60% showing clear granule cell morphology. We have now provided these numbers in text on page 7 in the manuscript.

References:

Abraham, H., Tornoczky, T., Kosztolanyi, G., and Seress, L. (2001). Cell formation in the cortical layers of the developing human cerebellum. Int J Dev Neurosci 19, 53-62.

Haldipur, P., Aldinger, K.A., Bernardo, S., Deng, M., Timms, A.E., Overman, L.M., Winter, C., Lisgo, S.N., Razavi, F., Silvestri, E., et al. (2019). Spatiotemporal expansion of primary progenitor zones in the developing human cerebellum. Science 366, 454-460.

Hatten, M.E. (1985). Neuronal regulation of astroglial morphology and proliferation in vitro. J Cell Biol 100, 384-396.

Lindholm, D., Dechant, G., Heisenberg, C.P., and Thoenen, H. (1993). Brain-derived neurotrophic factor is a survival factor for cultured rat cerebellar granule neurons and protects them against glutamate-induced neurotoxicity. Eur J Neurosci 5, 1455-1464.

Pibiri, V., Ravarino, A., Gerosa, C., Pintus, M.C., Fanos, V., and Faa, G. (2016).

Stem/progenitor cells in the developing human cerebellum: an immunohistochemical study. Eur J Histochem 60, 2686.

Selvadurai, H.J., Luis, E., Desai, K., Lan, X., Vladoiu, M.C., Whitley, O., Galvin, C., Vanner, R.J., Lee, L., Whetstone, H., et al. (2020). Medulloblastoma Arises from the Persistence of a Rare and Transient Sox2(+) Granule Neuron Precursor. Cell Rep 31, 107511.

Sutter, R., Shakhova, O., Bhagat, H., Behesti, H., Sutter, C., Penkar, S., Santuccione, A., Bernays, R., Heppner, F.L., Schuller, U., et al. (2010). Cerebellar stem cells act as

medulloblastoma-initiating cells in a mouse model and a neural stem cell signature characterizes a subset of human medulloblastomas. Oncogene 29, 1845-1856.

Wizeman, J.W., Guo, Q., Wilion, E.M., and Li, J.Y. (2019). Specification of diverse cell types during early neurogenesis of the mouse cerebellum. *eLife* 8.

[Editors' note: further revisions were suggested prior to acceptance, as described below.]

Your paper addresses the divergent mechanisms that create species-specific features of brain development, focusing on transcriptional programs, and their timing, that generate human cerebellar cells compared to those in mouse. You describe a rapid protocol for the derivation of the human ATOH1 lineage that generates excitatory cerebellar neurons from human embryonic stem cells (hESCs). You then developed a clonal hPSC line expressing EGFP under enhancer of ATOH1, enabling FAC sorting of these cells and further cell purification with an antibody against TAG1, expressed in postmitotic granule cells located in the premigratory zone, You successfully co-cultured these cells in a trans-well configuration with mouse glial cells, and observed migration of the human-derived cells after transplantation into the mouse cerebellum.Transcriptional profiling then indicated that the mouse postnatal ATOH1 lineage most closely matched the profile of the embryonic human cerebellum, with transcription factors classically associated with differentiated neurons in mouse situated in the human outer external granule layer housing granule cell precursors. Your results suggest mechanisms underlying the prolonged development of the cerebellum in the human that may be linked to its increased size in evolution.All three reviewers agreed that the manuscript has been greatly improved but there are some remaining issues that need to be addressed, as outlined below:Reviewer 1 asked for two further revisions:1. As NeuroD1 is expressed in other brain regions than cerebellum (including cortex and other forebrain regions cf for instance scrnaseq data on ucsc from Kriegstein lab) it would be very valuable to know the proportion of NeuroD1/Pax6 double positive cells (instead of each marker separately) as this combination is much more specific. Perhaps elaborate on your rebuttal comment #1.

Please see our response below as requested:

In the initial submission of the manuscript, we showed MAP2, NeuroD1, and Synaptophysin in addition to TAG1 expression in Tag1+ cells isolated at DIV28 and grown an additional 20 days in co-culture with mouse cerebellar neurons and glia. We also described the characteristic small round morphology of differentiated granule cells, which is not apparent when they are still proliferating or migrating. See Fig. 2E-F. In the brain, NeuroD1 expression is largely localized to cerebellar granule cells as well as in granule cells in the hippocampus and hence, it is a fairly selective granule cell marker. We have now performed additional analyses including the quantification of NeuroD1+ and Pax6+ cells at DIV28+20 and quantification of cells with a small round nucleus. The text on p.7 second paragraph has been updated with this information. We showed that the human cells expressed the GC markers NEUROD1 (42/56 examined, 75%, Fig. 2E) and PAX6 (174/218 examined, 80%). In response to the additional query by reviewer 1, regarding the cerebellar versus forebrain identity of the differentiated cells, we performed additional experiments to double label cells with PAX6 and NEUROD1 in the same cells, as suggested by the reviewer, and observed that 18/22 examined (81%) co-expressed both NEUROD1 and PAX6. Hence, cells that are PAX6+ tend to also be NEUROD1+ (the percentages of positive cells out of total human cells were similar in single and double labeling experiments and all cells that were NEUROD1 positive were also PAX6 positive in double labeling experiments). We have added a sentence on p. 7 line 25 about this. We would like to point out that these cells have already undergone a round of selection by way of TAG1-sorting and we previously showed that the great majority of TAG1+ cells co-express PAX6 (Fig. 2D, top panel). Moreover, we would like to bring the attention of the reviewers to Fig. 5A, *right*, where we show that the ATOH1 (TRAP) population at DIV28 co-expresses both *PAX6* and *NEUROD1*. Even though this is at the population level and at an earlier stage, it provides further support for the co-expression of these factors in the ATOH1 lineage derived by our method. Additionally, our analysis of patterning at DIV11 for markers of the cerebellar territory versus forebrain and midbrain territory showed co-expression of *GBX2* and *EN2* (mid/hindbrain) and lack of the fore/midbrain marker *OTX2* (Fig.1 -figure supplement 1A). Together, these data strongly suggest that the differentiated cells are cerebellar in identity. We have also added the expression of VGLUT1, a glutamatergic synaptic marker, expressed in cerebellar granule cell parallel fibers (see new Fig. 2G) as an additional marker of differentiated GCs.

To further address the question of the identity of cells negative for ATOH1 (~20% of the cells in cultures at DIV28), we carried out further double immunolabeling experiments at two timepoints:

1. At DIV28 using antibodies against PAX6, NeuN, Calretinin, and SOX2 (in combination with ATOH1-EGFP). We have evidence that the great majority of the cells are rhombic lip (RL) and RL derivatives. Evidence to support this conclusion includes a large number of cells that expressed PAX6 (which is expressed both in the EGL and the RL), see (Haldipur et al., 2019), a proportion of which are double positive for ATOH1-EGFP^+^;NeuN^+^ (see new Fig. 1 – Supplementary Fig. 3B). ATOH1 and NeuN are expressed in human GCPs (our finding) but are largely absent from RL cells (Haldipur et al., 2019) and hence regarding your question about the identity of ATOH1^-^ cells, our data suggest that they are mostly RL cells (PAX6+; NeuN-) with smaller contributions of differentiated GCs (PAX6+; NEUN+; ATOH1-EGFP-), cerebellar nuclei and/or Unipolar brush cells (Calretinin+, see new Fig. 1 figure supplement 3). We also examined SOX2 expression. SOX2 is typically expressed in the ventricular zone neuroepithelium, which is ATOH1^-^ and gives rise to the GABAergic lineage and glial progenitors. A small number of Sox2^+^ cells have been reported in the mouse EGL, although they constitute a rare population (Selvadurai et al., 2020; Sutter et al., 2010). Here, by immunohistochemistry, we show that a surprisingly large number of Sox2^+^ cells are present in the human EGL (Figure 5. Figure supplement 1D). Moreover, in our cultures, 72% +/-16 of the Sox2^+^ cells are ATOH1-EGFP+ (GCP identity), while 27% of the Sox2^+^ cells are EGFP-, suggesting that these cells (27%) may represent glial progenitors/ventricular zone progenitors. We have added a paragraph to page 6 to describe these data.

2. At DIV28+20, to estimate the proportion of cells that will go on to a unipolar brush cell or cerebellar nuclei identify (Calretinin+/NeuN+/-), we quantified the presence of these markers in the TAG1^-^ fraction after 20 days in culture post sorting (DIV28+20) and observed again that Calretinin+ cells constitute only a small fraction of the population (~5%). We have added a paragraph on Page 8 to describe these data.

Together, our analysis suggests that at DIV28 our cultures contain mostly RL cells and GCPs and a small number of cerebellar nuclei/unipolar brush cells, in addition to newly postmitotic GCs marked by Tag1 and PAX6+;NeuN+ (EGFP-/EN2+/-). We cannot exclude that some of the cells in culture may have brain stem identity. Hence purification steps such as Tag1 sorting at different time-points provides a strategy to obtain more pure populations of a particular cell type of interest.

2. In absence of more data on human time course of expression of NeuroD1 and other markers, the authors should tone down their conclusion on heterochrony – rather describe it as divergent pattern of expression that could be consistent with heterochrony.

We have now modified the text accordingly. The word “heterochronic” was deleted throughout. In some places it was replaced with a “shift” or “molecular divergence” or “divergent expression” throughout the manuscript (marked in red). Specifically, it was deleted in the abstract line 11, on p.3 line 26, on p. 9 line 25, on p.11 line 26 it was replaced by “molecular divergence”, on p.12 line 1 it was replaced by “divergent expression”

3. From the Reviewing Editor: with regard to differences in overall timing, it is most interesting that Sox2 is expressed in the human EGL unlike in mouse. You say in your rebuttal "we show that a surprisingly large number of Sox2^+^ cells are present in the human EGL (Figure 5. Figure supplement 1D). Moreover, in our cultures, 72% +/-16 of the Sox2^+^ cells are ATOH1-EGFP+ (GCP identity), while 27% of the Sox2^+^ cells are EGFP-…." Although you added a paragraph to page 6 to describe these data, it would be welcome to reiterate this in the Discussion to forify the idea that the cells in the human EGL are "held" in an extended immature (stem cell) state as well as heterochronically expressing transcription factors that in mouse are expressed at later stages.

Thank you for pointing this out. We have added a sentence to the discussion on p.12.